# Integrated modeling of the Nexin-dynein regulatory complex reveals its regulatory mechanism

Avrin Ghanaeian [1], Sumita Majhi [2], Caitlyn L. McCafferty[3], Babak Nami [4], Corbin S. Black [1], Shun Kai Yang [1], Thibault Legal [1], Ophelia Papoulas [3], Martyna Janowska[2,5], Melissa Valente-Paterno[1], Edward M. Marcotte [3], Dorota Wloga[2] ✉ & Khanh Huy Bui [1] ✉

Cilia are hairlike protrusions that project from the surface of eukaryotic cells and play key roles in cell signaling and motility. Ciliary motility is regulated by the conserved nexin-dynein regulatory complex (N-DRC), which links adjacent doublet microtubules and regulates and coordinates the activity of outer doublet complexes. Despite its critical role in cilia motility, the assembly and molecular basis of the regulatory mechanism are poorly understood. Here, using cryo-electron microscopy in conjunction with biochemical cross-linking and integrative modeling, we localize 12 DRC subunits in the N-DRC structure of *Tetrahymena thermophila*. We also find that the CCDC96/113 complex is in close contact with the DRC9/10 in the linker region. In addition, we reveal that the N-DRC is associated with a network of coiled-coil proteins that most likely mediates N-DRC regulatory activity.

Cilia are microscopic hair-like protrusions that extend from the surface of eukaryotic cells and are responsible for cell motility, sensory functions, and signaling. Ciliary motility has a critical role in the clearance of foreign particles from the respiratory system, establishment of the left-right asymmetry of the visceral body organs, development, and maintenance of the spinal cord, development and composition of the brain ventricular system, and fertilization in the reproductive tract[1].

The motile cilium has a highly conserved core axial cytoskeleton called the axoneme. It is a bundle of nine doublet microtubules (DMTs) composed of A- and B-tubules surrounding two singlet microtubules known as the central apparatus (Fig. 1a). Each DMT is composed of multiple repeating 96-nm units comprising identical sets of axonemal complexes such as outer and inner dynein arms (ODAs and IDAs), radial spokes and a nexin-dynein regulatory complex (N-DRC)[2–5] (Fig. 1b). Dynein arms are A-tubule-docked molecular motors that generate sliding force between DMTs[6–8]. ODAs generate high ciliary beat frequency, while IDAs control the cilia's bending shape[9,10]. Each 96-nm repeating unit contains four units of ODAs (two or three-headed, depending on the organism), one two-headed IDA (dynein I1/f), and six distinct single-headed IDAs (dyneins a–g) with different mechanical properties[3]. The N-DRC bridges neighboring DMTs and restricts sliding motions between DMTs. As a result, the N-DRC converts dynein-generated sliding motion into ciliary bending motion.

To generate the oscillating waveform movement of cilia, dynein activities switch back and forth from one side to the other side of the axoneme[11]. Such dynein regulation is accomplished through coordinated activities of mechano-regulatory complexes, including the central apparatus, radial spokes, N-DRC, and CCDC96/113 complexes[12]. Regulatory signals originating at the central apparatus are transmitted through the radial spokes and other mechano-regulatory complexes on the DMT to the dynein arms. Signal transmission mechanisms, however, remain unclear.

[1]Department of Anatomy and Cell Biology, Faculty of Medicine and Health Sciences, McGill University, Québec, Canada. [2]Laboratory of Cytoskeleton and Cilia Biology, Nencki Institute of Experimental Biology of Polish Academy of Sciences, 3 Pasteur Street, 02-093 Warsaw, Poland. [3]Department of Molecular Biosciences, Center for Systems and Synthetic Biology, University of Texas, Austin, TX, USA. [4]Genetics and Genome Biology Program, Hospital for Sick Children, Toronto, Canada. [5]Present address: Laboratory of Immunology, Mossakowski Institute of Experimental and Clinical Medicine, Polish Academy of Science, Pawinskiego 5, 02-106 Warsaw, Poland. ✉e-mail: d.wloga@nencki.edu.pl; huy.bui@mcgill.ca

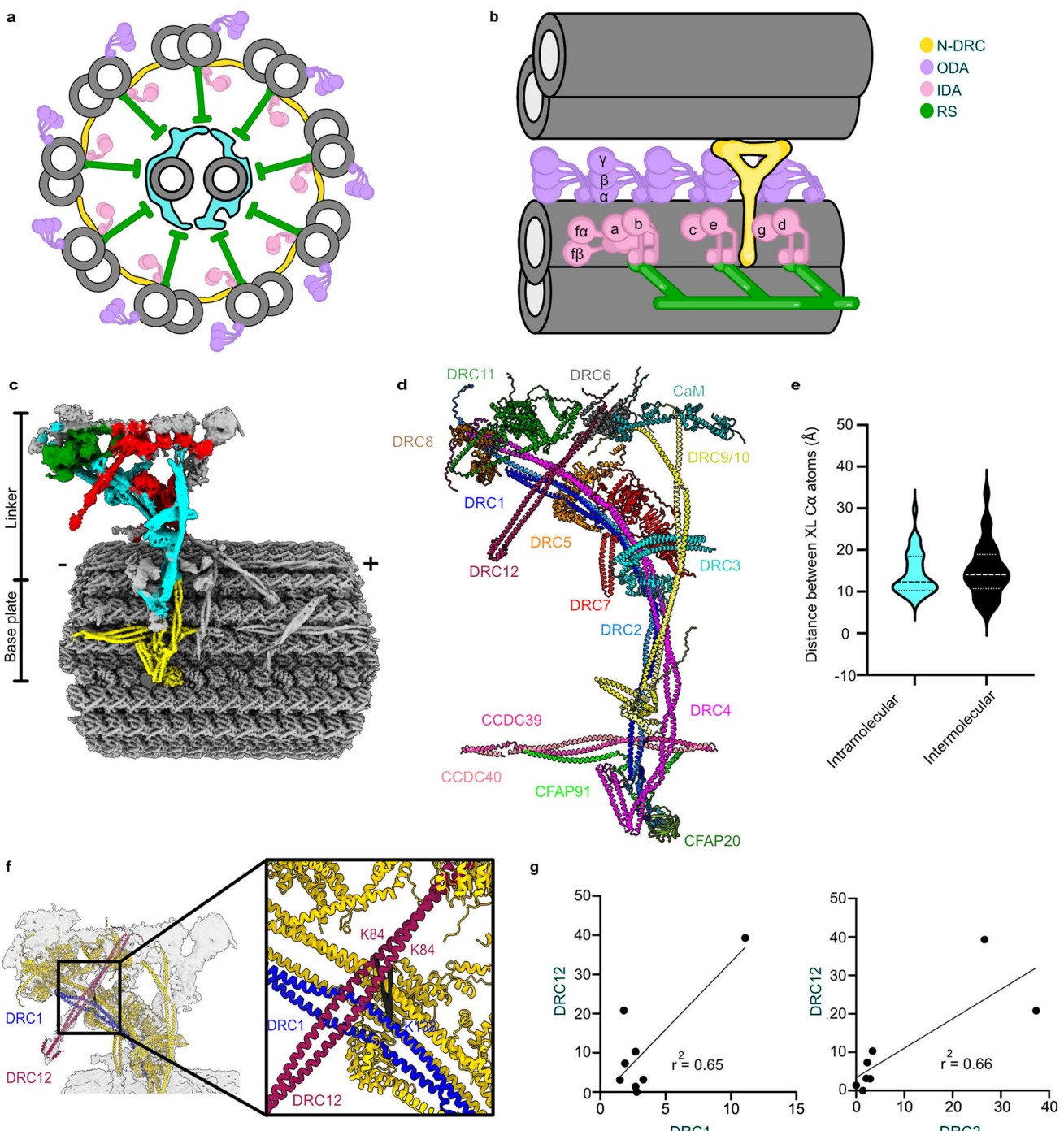

**Fig. 1 | An overview of the structure of cilia and N-DRC. a** Cross-sectional view of a cilium. **b** Longitudinal view of the 96-nm repeating unit of a DMT and its adjacent DMT connected by the N-DRC (yellow). **c** The composite cryo-EM map of the N-DRC and DMT. The colors of different regions indicate the modeling methods. Yellow indicates the base plate part which was modeled using Coot. The cyan, red, and green colors represent the regions modeled using AlphaFold, AlphaFold and cross-links, AlphaFold and Assembline, respectively. The remaining gray color in the top of the linker region represents unidentified regions. Signs (+) and (−) indicate the distal and proximal ends of the DMT. **d** The pseudoatomic model of the N-DRC. **e** Violin plots of the distance between intermolecular and intramolecular Cα atoms of chemically cross-linked residues mapping on the N-DRC model.

The intramolecular and intermolecular cross-links are shown in cyan and black, respectively. A maximum distance of 35 Å is expected for the DSSO cross-links. **f** Localization of DRC12 using a cross-link with DRC1. The lengths of the cross-link between DRC1 and each copy of DRC12 are 27.26 and 30.61 Å. **g** Correlation graphs of consensus normalized mRNA expression levels shown as transcripts per million (TPM) of total mRNA for two selected pairs of genes (DRC12/DRC1 and DRC12/ DRC2). Pearson R and two-tailed *P* values were calculated by applying simple linear regression and F test to examine significance of correlation ratios. *P* value for DRC1 and DRC12, and DRC2 and DRC12 were 0.0148 and 0.0141, respectively. *P* value < 0.05 was considered as significant correlation.

**Table 1 | Cryo-EM data collection and refinement parameters for all datasets used in this study**

| Method | Single-particle analysis | | | | Subtomogram averaging |
|---|---|---|---|---|---|
| Dataset | *WT* | *SB255* | *MEC17* | *K40R* | *WT* |
| Microscope | Titan Krios | Titan Krios | Titan Krios | Titan Krios | Titan Krios |
| Electron detector | Gatan K3 | Gatan K3 | Gatan K3 | Gatan K3 | Gatan K3 |
| Zero-loss filter (eV) | 30 | 30 | 30 | 30 | 20 |
| Magnification | 64,000 | 64,000 | 64,000 | 64,000 | 42,000 |
| Voltage (kV) | 300 | 300 | 300 | 300 | 300 |
| Exposure (e/A²) | 45 | 45 | 45 | 45 & 73 | 160 |
| Defocus (µm) | 1.0–3.0 | 1.0–3.0 | 1.0–3.0 | 1.0–3.0 | 2.5–3.5 |
| Pixel size (Å) | 1.37 | 1.37 | 1.37 | 1.37 | 2.12 |
| Tilt range (increment) | – | – | – | – | −60° to 60° (3°) |
| Tilt scheme | – | – | – | – | dose symmetric |
| Movies acquired | 18,384 | 25190 | 6545 | 25,610 | – |
| Particle number | 78,310 | 54,489 | 6989 | 71,714 | – |
| Tilt series acquired | – | – | – | – | 58 |
| Subtomo averaged | – | – | – | – | 2608 |
| Symmetry imposed | C1 | C1 | C1 | C1 | C1 |
| Map resolution (Å) | 3.60 | | | | 19 |

The N-DRC was first discovered in electron microscopy (EM) images of axonemal cross-sections from *Chlamydomonas reinhardtii* and described as a nexin link that connects two adjacent DMTs[13]. Later, genetic studies revealed the presence of a "dynein regulatory complex" (DRC). Mutations in some *drc* subunit genes in *C. reinhardtii* rescued paralysis caused by mutations in radial spokes and central apparatus proteins[14–16]. Applications of cryo-electron tomography (cryo-ET) finally revealed that the DRC and the nexin link are the same structure and coined the term "N-DRC"[17].

The N-DRC is a large 1.5-MDa, ~50-nm-long Y-shaped complex with two main regions: the base plate and linker regions (Fig. 1c). Its base plate attaches to the inner junction of the DMT and four A-tubule protofilaments. Its linker region projects away through the inter-DMT space and connects A-tubule to the neighboring B-tubule[17]. To date, biochemical and genetic studies have revealed that the N-DRC is composed of 11 subunits in *C. reinhardtii* and other species[18]. Cryo-ET and subtomogram averaging of wild-type and N-DRC mutants of *C. reinhardtii* revealed that DRC1, DRC2, and DRC4 span through the N-DRC from the base plate to the linker region, while DRC3, DRC5-11 were localized in the linker[17,19–21]. A high-resolution cryo-electron microscopy (cryo-EM) study of the DMT from *C. reinhardtii* revealed that DRC1, DRC2, and DRC4 form the core structure of the base plate[22]. Mutations in DRC subunit genes of *C. reinhardtii* caused the disassembly of the N-DRC, defects in ciliary movement and destabilized the assembly of several closely associated structures, such as the IDAs[9,23–25].

Despite extensive research on the N-DRC, the precise composition and structural model of the components has yet to be determined. Consequently, our understanding of the intra- and inter-interactions of the N-DRC and its function in dynein regulation and ciliary movement is still fragmentary. Here, we use integrated modeling and cryo-EM to model the entire N-DRC structure and some associated proteins from the ciliate *Tetrahymena thermophila*. Our study reveals 12 N-DRC components and their detailed organization. Our structural model also shows that the N-DRC regulates ODAs and IDAs by interacting with a network of coiled-coil proteins.

## Results

### Integrated modeling reveals the molecular architecture of the N-DRC

To get the N-DRC structure, we performed a single-particle analysis of isolated DMT from *T. thermophila* from several datasets (Table 1 and

"Methods"[26,27]. We obtained the 96-nm repeating unit of the DMT at a global resolution of 3.6 Å (Supplementary Fig. 1), containing the N-DRC base plate. This allowed the detailed modeling of the N-DRC base plate, which comprises segments of DRC1, DRC2, DRC4A, DRC4B, and associated proteins on the surface of the DMT, CCDC39, CCDC40, and CFAP91 (Supplementary Fig. 2a and Supplementary Table 1).

However, the N-DRC linker region was not well resolved because of its flexibility. Therefore, we developed an image processing strategy to obtain the linker structure at ~5–10 Å resolution involving extensive subtractions, three-dimensional classifications, and focused refinement (Supplementary Fig. 1). This resolution suffices to fit the AlphaFold2 predicted structures of N-DRC components unambiguously, especially the coiled-coil part of the proteins (Fig. 1c, d and Supplementary Fig. 2a–f, and Supplementary Tables 1 and 2). As a result, we can confidently fit the remaining AlphaFold2 multimer model of DRC1/2, DRC4A/4B, DRC9/10 complexes, DRC3 and DRC5 into the N-DRC linker region (Supplementary Fig. 2c–e).

For the remaining regions where we could not fit in the predicted structures uniquely, we performed integrated modeling using the Assembline program and spatial restraints derived from cross-links identified by in situ cross-linking mass spectrometry of the *T. thermophila* cilia[28]. We localized DRC11A and two copies of DRC8 to the proximal lobe of the linker and DRC7 in the middle part of the linker (Fig. 1c, d). Integrated modeling yields two possible positions for DRC6A (Supplementary Fig. 3a). Overall, our integrated model of the N-DRC and associated proteins explains ~85% of the analyzed density (Fig. 1c, d, Table 2, and Supplementary Movie 1). However, we could not localize proteins to the N-DRC distal lobe.

Next, we used the intermolecular and intramolecular and intramolecular cross-links between N-DRC components and associated proteins from in situ cross-linking mass spectrometry to validate our model (Supplementary Tables 3 and 4). Our model satisfies all the spatial restraints from cross-links found for N-DRC proteins (Fig. 1e and Supplementary Fig. 2b). Interestingly, we found a cross-link from a conserved protein, CCDC153 (UniProtID Q22RH5), to DRC1 in the linker region (Fig. 1f and Supplementary Table 3). The density of the unknown coiled-coil from the N-DRC linker region matches the AlphaFold2 predicted structure of the CCDC153 homodimer (Fig. 1f). We propose CCDC153 is an N-DRC subunit that we named DRC12. To verify DRC12 localization, we performed BioID assays using DRC1-HA-BirA* and DRC2-HA-BirA* as baits. Most DRC components were

**Table 2 | Refinement statistics of the base plate and linker models**

| | Base plate | Linker | Full model |
|---|---|---|---|
| Model-to-Map fit, CCmask | 0.43 | 0.56 | 0.56 |
| All-atom clashscore | 17.70 | 102.53 | 107.97 |
| **Ramachandran plot** | | | |
| Outliers [%] | 0.11 | 0.08 | 0.17 |
| Allowed [%] | 3.45 | 7.66 | 6.31 |
| Favored [%] | 96.44 | 92.26 | 93.52 |
| Rotamer outliers [%] | 1.00 | 0.00 | 0.01 |
| Cbeta deviations [%] | 0.06 | 0.06 | 0.01 |
| Cis-proline [%] | 0.00 | 1.8 | 3.3 |
| Cis-general [%] | 0.1 | 0.1 | 0.1 |
| Twisted proline [%] | 3.7 | 1.8 | 0.7 |
| Twisted general [%] | 0.1 | 0.4 | 0.2 |

biotinylated in both experiments, including DRC12 (Supplementary Table 5), validating the proximity of DRC1 and DRC2 to DRC12. The high correlation of the normalized RNA expression of CCDC153 from different human tissues with that of DRC1 (CCDC164) and DRC2 (CCDC65) (Fig. 1g) and other N-DRC components (Supplementary Fig. 4a) further supports the assignment of CCDC153 as a DRC protein.

Interestingly, there is a cross-link between DRC12 and an F-box domain-containing protein (UniProtID Q24CL2) (Supplementary Table 3), which is the fold observed in DRC6A and DRC6B (Supplementary Fig. 3b). Coincidentally, the cross-link would put the protein at the center of the linker region, overlapping with one position of DRC6A determined by integrated modeling. Therefore, we named the F-box protein DRC6C. Out of the three paralogs, DRC6C is the most abundant, suggesting that DRC6C represents in most of the N-DRC structures in the DMT. We have identified a cross-link between DRC6C and a calmodulin protein (UniProtID I7MDA9) that corresponds to the small globular densities observed in our map (Fig. 1d and Supplementary Fig. 4b). Our BioID results, using DRC1-HA-BirA*, DRC2-HA-BirA*, and DRC3-HA-BirA* as baits, have demonstrated the biotinylation of both calmodulin and DRC6C (Supplementary Fig. 4c–e and Supplementary Table 5). This validation provides compelling evidence for the presence of DRC6C and calmodulin within the N-DRC. In our model, we used two calmodulin molecules, since the densities in the distal lobe are explained better when using two instead of one calmodulin (Fig. 1c, d). Each calmodulin molecule was predicted to bind to DRC9 and DRC10 by AlphaFold2 Multimer.

DRC4 forms a homodimer in *C. reinhardtii*[22]. *T. thermophila* contains two DRC4 paralogs, DRC4A and DRC4B. The identified intermolecular cross-links between DRC4A and DRC4B, but not intramolecular cross-links, suggest that DRC4A and DRC4B predominantly form a heterodimer (Supplementary Tables 3 and 4). To further support that DRC4A and DRC4B exist as heterodimers, we performed pull-down assays using anti-GFP beads and DRC4A-GFP or DRC4B-GFP as bait. Both proteins efficiently pulled down the HA-tagged DRC4 paralog, DRC4A-GFP pulled down DRC4B-HA and vice versa, DRC4B-GFP interacted with DRC4A-HA, while homodimeric interactions between DRC4 paralogs were poor or undetectable (Supplementary Fig. 5a, b). Interestingly, the pull-down analysis indicated that DRC3-HA also interacted with DRC4B, but poorly with DRC4A-GFP. Further analysis of the AlphaFold2 multimer prediction of DRC4A/4A, DRC4B/4B, and DRC4A/4B dimers shows that only the DRC4A/4B heterodimer matches the DRC4 density in our map (Supplementary Fig. 5c, d). These results imply that in *T. thermophila*, the DRC4 dimer exists mainly as a DRC4A/4B heterodimer (Supplementary Fig. 5c, d) (see "Methods").

## The N-DRC core scaffold consists of coiled-coil proteins

The N-DRC backbone comprises coiled-coil proteins (DRC1, DRC2, DRC4A, DRC4B) that extend from the base plate to the linker region (Fig. 1d). These coiled-coil proteins can give the N-DRC elasticity.

In *T. thermophila* cilia, the C-terminal fragments of the DRC1/2 and DRC4A/4B heterodimers are located on the A-tubule and form the N-DRC base plate (Fig. 1d and Supplementary Fig. 2a), similar to *C. reinhardtii*[22]. At the base plate, the DRC1/2 and DRC4A/4B heterodimers contact perpendicularly with the coiled-coil CCDC39/40, the "molecular ruler" that runs along the DMT in the wedge of protofilaments A2A3 and determines the 96-nm repeat of the DMT. The C-terminal region of the DRC1/2 heterodimer also interacts with (i) the C-terminal region of the DRC4A/B heterodimer, (ii) CFAP91, which runs almost parallel to the CCDC39/40 coiled-coil (Fig. 2a) and (iii) CFAP20, an inner junction protein[27,29]. Previous studies showed CFAP91 extends from the base of RS2 through the N-DRC base plate to the radial spoke RS3, playing a significant role in stabilizing and localizing radial spokes RS2 and RS3 on the DMT[22,30,31]. N-DRC positioning on the DMT may depend on interactions between the DRC1/2 coiled-coil and the DRC4A/B coiled-coil and the CCDC39/40 coiled-coil (Fig. 2a).

The N-DRC core scaffold is composed of DRC1, DRC2, and DRC4, which interact with DRC3, DRC5, DRC8, and DRC11 in the linker region (Fig. 2a–c). Our model indicates that the DRC1/2 coiled-coil interacts with the DRC4A/B coiled-coil not only in the base plate region but also in the top part of the linker (Fig. 2a–c). This coiled-coil bundle constructs the N-DRC core structure (Fig. 1d). Therefore, the DRC1/2 and DRC4A/B complex is critical for the assembly of the N-DRC since they position the N-DRC base plate to the DMT via interaction with CFAP20 in the inner junction and interact with all the proteins in the N-DRC central and proximal lobes. This is consistent with the phenotypes of the *C. reinhardtii* mutants of *DRC1* (*pf3*) and *DRC2* (*ida6*), and human mutations in the *CCDC164* (*DRC1*) and *CCDC65* (*DRC2*) genes, which cause the complete loss of the N-DRC[24,25,32]. As an N-DRC core scaffold component, the loss of the DRC4 protein in the *pf2* mutant in *C. reinhardtii* causes the absence of all DRC subunits except the DRC1/2 complex[15,18,24].

DRC9/10 is another coiled-coil dimer that binds to the DRC1/2 coiled-coil just above the base plate. The dimer projects through the linker region distally to form the other arm of the Y-shape of the N-DRC and interact with DRC3 in the distal region (Figs. 1d and 2c).

Within the N-DRC linker region, the leucine-rich DRC5 subunit interacts extensively with DRC1/2 and DRC4A/4B coiled coils (Fig. 2b). In the top part of the linker, the extended N-terminus of DRC2 interacts with DRC8, while DRC1 folds back to the central part of the linker and associates with DRC5 (Fig. 2b and Supplementary Fig. 2c). The DRC4A/B heterodimer plays an essential role in the stability of the N-DRC by interacting with the leucine-rich domains of DRC3, DRC5, and DRC7 in the central part of the linker and interacting with DRC11 in the linker top part (Fig. 2b, c).

DRC7 is in the middle part of the linker, which connects the N-DRC proximal and distal lobes. In the N-DRC distal lobe, DRC7 interacts with the DRC9/10 coiled-coil and the α-helix bundle domain of DRC3. In the proximal lobe, DRC7 interacts with DRC4A/4B coiled-coil (Fig. 2c). Therefore, DRC7 can be a signal transfer bridge between the distal and proximal lobes. Although DRC7 connects the distal domain to the proximal lobes by linking DRC4A/B to DRC10, loss of DRC7 in the *drc7* mutant does not affect the assembly of the distal subunits (DRC3, DRC9, and DRC10)[21], suggesting that DRC7 likely reinforces the distal lobe but is not essential for the assembly of the distal lobe.

From AlphaFold2 prediction and density fitting, our model predicted that two copies of EF-hand DRC8 proteins bind tightly to the helical hairpin domain of the AAA + DRC11 protein (Fig. 2b), agreeing with the localization of DRC11 shown using *C. reinhardtii*[21] and *Trypanosoma brucei*[33] mutant strains. Interestingly, superimposition of the subtomogram averaged map of human DMT (EMD-5950)[34] onto that of

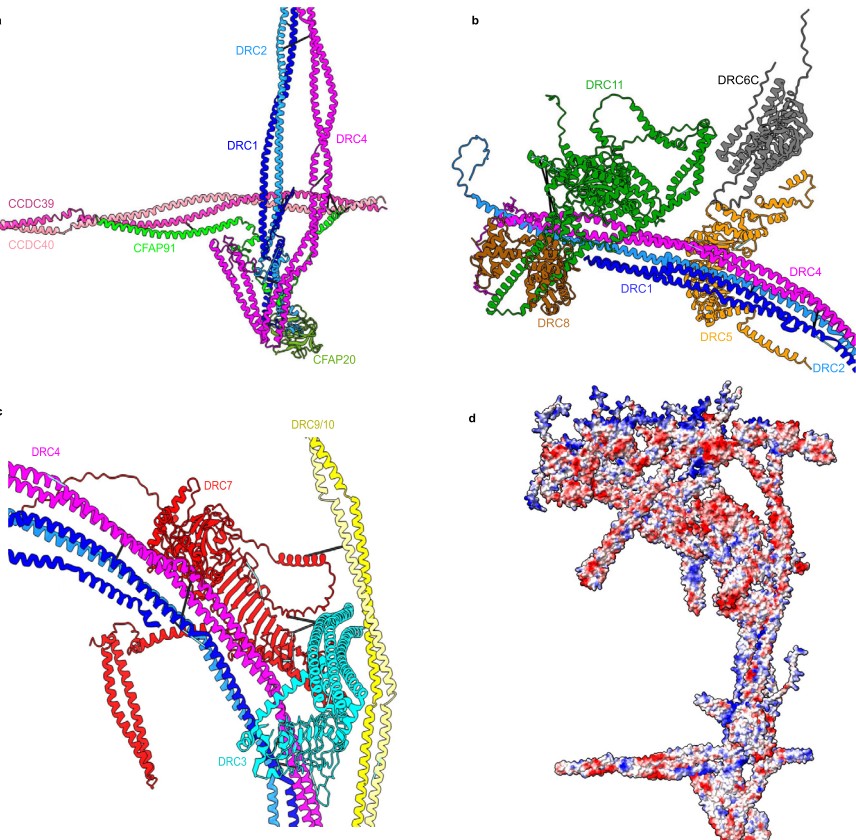

**Fig. 2 | Interactions within the N-DRC. a** The interactions in the base plate region. **b** DRC5 and DRC11/8 subcomplex interact with scaffold proteins (DRC1, DRC2, and DRC4). DRC12 and DRC7 are hidden for clarity. **c** The interactions between proteins in the N-DRC distal lobe. DRC7 interacts with the DRC9/10 coiled-coil and DRC3 in the linker distal part and with DRC4A/4B coiled-coil in the proximal part. The cyan and black lines represent intramolecular and intermolecular cross-links mapping onto the model, respectively. **d** The electrostatic surface charge of the N-DRC model. The loops at the top of the linker (DRC11A, DRC6C, DRC2, and DRC9/10), which connect to the adjacent DMT, are positively charged. The blue, red, and white colors indicate positive, negative, and neutral charges, respectively.

*T. thermophila* and AlphaFold2 Multimer modeling of the N-DRC in humans revealed that only one copy of DRC8 is present in the human N-DRC (Supplementary Fig. 5e, f). DRC11 contains an AAA+ domain that potentially contributes to conformational changes in the N-DRC[18]. In addition, DRC11 has a regulatory role since the knockdown of DRC11 homolog (CMF22) in *T. brucei* causes a motility defect[35].

The electrostatic surface charge of our N-DRC model indicates that the top of the linker region contains positively charged flexible loops (Fig. 2d). This allows the binding of the N-DRC to the negatively charged tubulin from the adjacent DMT[36]. This also serves as validation for our N-DRC model, since we are not using surface charge information when building it.

**The CCDC96/113 complex interacts with the N-DRC**

In our cryo-EM map, the CCDC96/113 heterodimer, the conserved mechano-complex distal to the N-DRC[12], resolves well in the basal part, allowing de novo modeling of this region. To observe better the interaction of the entire CCDC96/113 structure to the N-DRC, we also performed cryo-ET and subtomogram averaging to obtain the 96-nm repeat of *T. thermophila* at 20 Å resolution (Supplementary Fig. 6a, b). The C-terminal ends of CCDC96 and CCDC113 form a clear coiled-coil domain (Fig. 3a). AlphaFold2 multimer prediction of CCDC96/113 is highly confident and accurately represents the topology of the coiled-coil in the cryo-EM map, allowing unambiguous docking of the CCDC96/113 heterodimer in the region away from the DMT surface (Supplementary Fig. 7a). The coiled-coil domain of CCDC96/113 is located parallel to the N-DRC, but from amino acid 458 onward, it forms a 90-degree turn toward the N-DRC distal part (Fig. 3a). The

N-terminus of CCDC96 in *T. thermophila* is not conserved and contains an extra WD40 beta-propeller compared to humans and *C. reinhardtii* (Supplementary Fig. 7b–d). Following the 90-degree turns of CCDC96, we predict that the non-conserved globular domain of CCDC96 extends to the N-DRC distal lobe by a flexible loop and localizes close to DRC10 (Fig. 3a). We compared the subtomogram average of *T. thermophila* wild-type DMT to that of *T. thermophila* CCDC96 deletion (EMD-12121)[12] and *C. reinhardtii* (EMD-20338)[21]. Apparently, the density of the distal lobes of *C. reinhardtii* is significantly thinner than that of *T. thermophila*, suggesting that the N-terminus of CCDC96 is there (Supplementary Fig. 7b). In addition, the WD40 domain of the N-terminus of CCDC96 is missing in the cryo-ET map of the *T. thermophila* CCDC96 deletion (Supplementary Fig. 7b). The sub-tomogram average of the 96-nm repeating unit also shows that the CCDC96/113 complex interacts with the base part of RS3 (Fig. 3b), as shown by a previous study[12].

As suggested by previous studies[25,37], our results also showed that the N-termini of DRC9 and DRC3 interact with dynein e and g, respectively, suggesting that they are regulatory hubs for such dynein (Fig. 3c).

We also found a cross-link from CFAP337A (UniProtID I7MM07, paralog CFAP337B UnitProtID I7MKT5) to the coiled-coil region of CCDC96. The AlphaFold2 predicted models of CFAP337A and CFAP337B contain two WD40 beta propellers and a bundled coiled-coil domain that fits unambiguously in a density close to the upper region of CCDC96 (Supplementary Fig. 8a). Mass spectrometry analysis confirmed that both CFAP337A and CFAP337B are lost in CCDC96 knockout cells[12]. Fitting of a predicted CFAP337 model, combined with

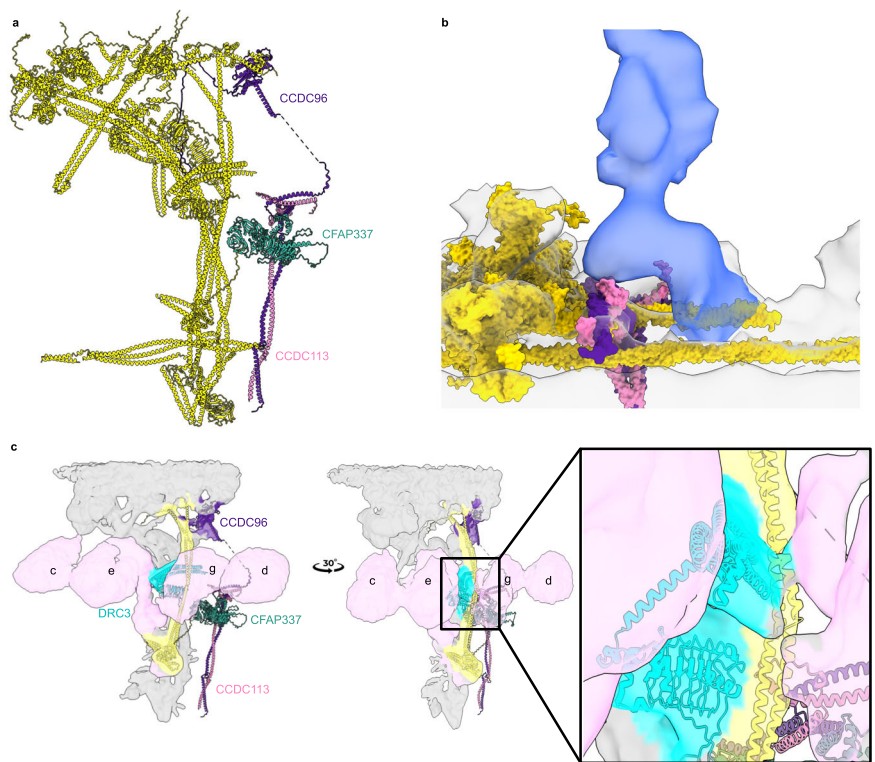

**Fig. 3 | Structure of the CCDC96/113 complex. a** The C-terminus of the CCDC96/113 coiled-coil is located on the DMT, and the N-terminus is localized near the N-DRC linker region. Colors: N-DRC proteins and coiled-coil associated proteins (CCDC39/40, CFAP91 and CFAP20), yellow; CCDC96, dark purple; CCDC113, pink; CFAP337, turquoise. **b** The C-terminus of the CCDC96/113 complex (dark purple and pink) interacts with the base of RS3 (blue density). **c** The interactions of DRC3, DRC9/10 with IDAs (light pink), and CCDC96/113 are demonstrated in the cryo-EM map and model.

cross-links and mass spectrometry data of the CCDC96-deletion mutant, demonstrated with high confidence that CFAP337 interacts directly with CCDC96. Our subtomogram average showed that the WD40 domain of CFAP337 mediates interaction with the stem of dynein d (Supplementary Fig. 8b). Therefore, CFAP337 might be an IDA regulatory hub or linkage.

### The N-DRC regulates dynein activity through direct IDA interactions and a network of coiled-coil proteins that spans the 96-nm repeat

The next step is to determine how the N-DRC regulates dynein activity and ciliary bending. To address this, we analyzed the single-particle and subtomogram averaged maps looking for any potential linkage from the N-DRC to the adjacent DMT and other axonemal complexes.

There are two observable connections from the N-DRC to the adjacent DMT, a clear connection from the distal lobe to protofilament B9 and a thin density from the middle of the linker region to protofilament B8 (Fig. 4b, black arrows). Based on our model, the density that connects to B8 corresponds to DRC6 (Supplementary Fig. 8c). Previous studies showed that $Sup_{pf}4$ mutation, which lacks DRC5 and DRC6 rescue the paralysis caused by defect in radial spoke[15]. Therefore, our structure suggests that the connection of DRC6 to the adjacent DMT can act as a brake for dynein arms. The rest of the N-DRC linker region does not show clear connections with the adjacent DMT and maintains a gap of ~25 Å to protofilament B9. It is possible that the disordered regions of DRC2, DRC8, DRC9, DRC10, DRC11, and DRC12 can still contact the highly negative and flexible tails of tubulins from protofilament B10 of the neighboring DMT (Fig. 2d).

We fitted atomic models of dyneins to the ODA and IDA densities (Fig. 4b and Supplementary Fig. 8d). The stalks of ODA α, β, and γ dyneins contact protofilaments B5, B6, and B7, respectively, as reported previously[38] (Fig. 4c). Regarding IDA, the stalks of single-headed dyneins a, b, c, d, e, and g all bind to protofilament B10, while the stalks of double-headed inner arm dyneins fα and fβ bind to protofilament B9 (Supplementary Fig. 8e, f). Interestingly, the stalk of dynein e appears to be in contact with the N-DRC proximal lobe corresponding to DRC8 and DRC11 (Fig. 4d). While the N-DRC and the stalk of dynein c do not seem in contact in the subtomogram average representing the postpower stroke state of dynein, the stalk of dynein c in the prepower stroke will move distally[39] (Fig. 4b, e) and therefore could be in contact with the N-DRC bulky proximal lobe. Besides physical contact from DRC3 to dyneins e and g described in the previous section, we showed that the activity of the stalks of dyneins c and e can be regulated by the N-DRC linker region, particularly the DRC8/11 complex, through mechanical contact. Dynein c is a strong dynein and is particularly needed for cell movement at high viscosity[40] The swimming speed of the *C. reinhardtii* dynein c mutant (*ida9*) is only marginally reduced compared to wild-type cells in a normal medium but greatly reduced in viscous media[40]. Similarly, the *C. reinhardtii* DRC11 mutant (*drc11*) swims only slightly slower than wild-type cells in normal medium[21]. Therefore, it might be possible that missing DRC11 affects dynein c regulation and has the same effect as missing dynein c.

Next, we sought complexes that are in contact with the N-DRC. We identified a coiled-coil density on the A3 and A4 protofilaments (Fig. 5a). This density extends through the CCDC96/113 complex and interacts directly with the WD40 domain of CFAP337. In *C. reinhardtii*, knockout of CFAP57 leads to the loss of CFAP337[41], suggesting that CFAP57 is a strong candidate for the A3A4 coiled-coil. There are four CFAP57 paralogs in *T. thermophila*, with CFAP57A and CFAP57C (UniProtID Q234G8 and W7WWA2) as the most abundant paralogs (Supplementary Table 2). We found two cross-links between CFAP57A and CFAP57C. The AlphaFold2 multimer predictions of CFAP57A and CFAP57C match well with the density in our map in both the C- and N-terminal regions (Supplementary Fig. 8g) and localization by cryo-ET

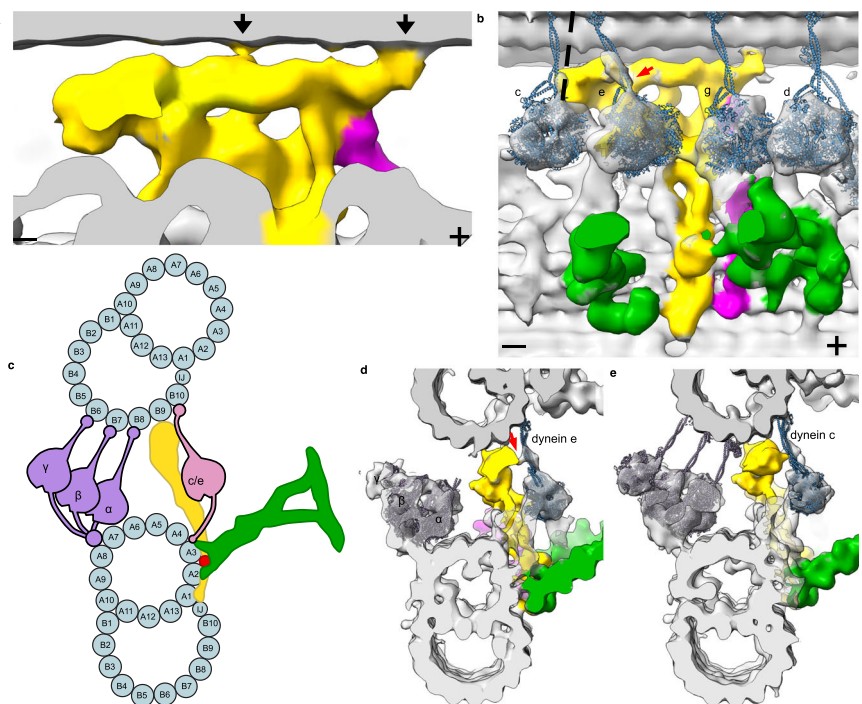

**Fig. 4 | Interactions between the N-DRC and neighboring MTD. a** Contacts of the N-DRC with the adjacent DMT are indicated by black arrows. Signs (+) and (−) indicate the distal and proximal ends of the DMT. Colors: N-DRC, yellow; Radial spoke, green; CCDC96/113, dark purple. **b** Fitting of dynein heavy chain models into the IDA densities shows that the dynein e stalk interacts directly with the N-DRC proximal lobe (red arrow), while the dynein c stalk in the prepower stroke conformation (dotted line) might interact with the N-DRC proximal lobe. **c** A cross-sectional model of how dyneins and the N-DRC contact the neighboring DMT. **d** Cross-sectional view of the interaction between dynein e and the N-DRC. The same connection in (**c**) is indicated by the red arrow. **e** Cross-sectional view of the potential interaction between dynein c and the N-DRC.

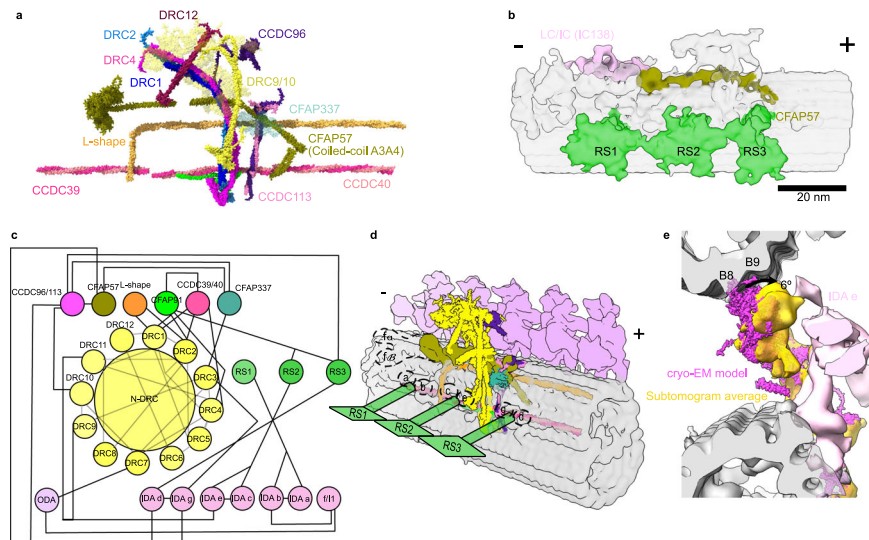

**Fig. 5 | The coiled-coil network associated with the N-DRC. a** The N-DRC in complex with the coiled-coiled protein network. **b** The location of CFAP57 dimer (dark green) shown in the 96-nm repeat subtomogram average of the DMT (IDAs and ODAs are hidden for better visualization). Scale bar, 20 nm. **c** Diagram of the N-DRC interactions with the coiled-coil network and IDAs and ODAs. **d** Schematic view of the 96-nm repeating unit and the coiled-coil network. The dashed shapes indicate the IDA locations along the 96-nm repeat. Signs (+) and (−) indicate the distal and proximal ends of the DMT. **e** Cross-sectional view of the N-DRC model from the cryo-EM map and in the subtomogram average of the DMT from intact cilia.

from a previous study[41]. CFAP57A was highly biotinylated in cells expressing CCDC96-HA-BirA* or CCDC113-HA-BirA* fusion proteins[12]. This supports that the A3A4 coiled-coil is CFAP57. CFAP57 spans almost 96 nm along the DMT[41] with the N-terminal region connecting to the I1 intermediate chain/light chain domain and the C-terminus positioned proximal to the radial spoke RS1 in the next 96-nm repeat (Fig. 5b). Our subtomogram average and single-particle maps suggest that the C-terminal region of CFAP57 also interacts with RS3 (Supplementary Fig. 8h) and that the CFAP57A/C dimer extends past CFAP337 through the N-DRC and toward the I1 intermediate chain/light chain complex. Our structural data showed that the CFAP57 heterodimer mediates interaction with the base of dyneins g and d (Supplementary

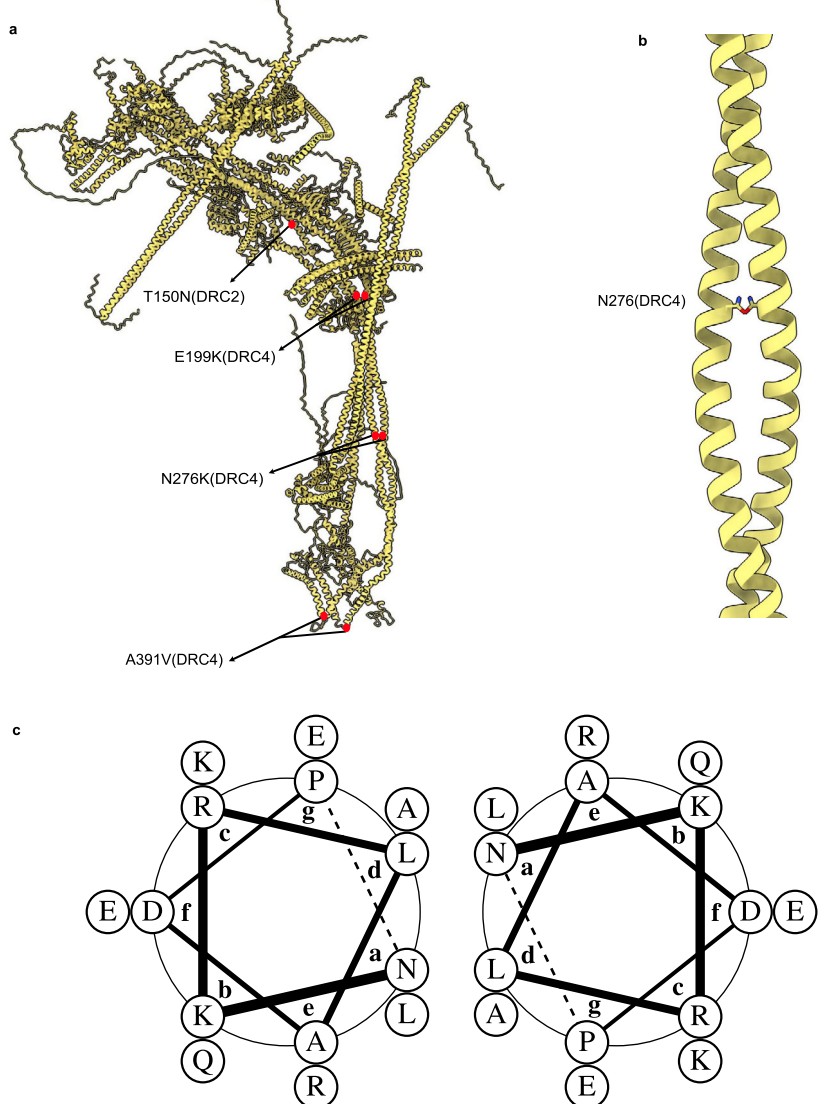

**Fig. 6 | Genetic mapping of missense ciliopathy causing mutations in the human N-DRC model. a** Red dots indicate missense mutations in DRC4 and DRC2. **b** Interaction of the side chain of asparagine 276 residues in DRC4 coil coiled. **c** Helical wheels of residues 276–289 segment of DRC4 coiled-coil (sequence: NKRLADPLQKAREE) showing the interaction of asparagine 276.

Fig. 8j). In CFAP57-KO cells, dyneins g and d are decreased compared with those in WT cells[41,42]. Therefore, CFAP57 can be an adaptor protein for the assembly of dynein g and d. This protein can be a critical regulatory hub for dyneins d and g since this coiled-coil interacts directly with CCDC96/113, CFAP337, and the N-DRC.

Our cryo-EM map also reveals the conservation of the L-shaped density discovered first in *C. reinhardtii* DMT[22] (Fig. 5a). The L-shaped coiled-coil is located between RS1 and RS2 on protofilament A1 and extends to the junction of protofilaments A4 and A5. The L-shaped coiled-coil interacts with CCDC39/40, DRC1/2, CCDC96/113 complexes, and CFAP337 (Fig. 5a, c, d). Overall, the N-DRC interacts with a network of coiled-coil proteins that can play a crucial role in regulating multiple dyneins.

Our subtomogram average does not show a linker density between the N-DRC and the ODA (Supplementary Fig. 8b), as shown in *C. reinhardtii*[21]. In contrast to *T. thermophila*, DRC7 in *C. reinhardtii* has extra LRR and Kelch domains at its N-terminus. Therefore, the observed linker density can be the LRR and Kelch domains of CrDRC7. However, the flexible loop of TtDRC7 (N-terminal) might still interact with the IC/LC tower of ODA for N-DRC-ODA regulation.

Next, we want to see whether the N-DRC adopts different states between the relaxed conformation in the isolated DMT and the in situ conformation in the cilia. Interestingly, when overlapping the DMT from the cryo-EM map of isolated DMT to the DMT of the subtomogram average, the linker region in the subtomogram average has a 6° tilt toward the inner dynein arms (Fig. 5e). As a result, the N-DRC interacts with the B9 protofilament in situ instead of the B8 protofilament. The in situ conformation might link to the interaction between the positively charged linker region of the N-DRC and the specific polyglutamylation of B9 protofilament[43]. As the conformational change observed in the N-DRC linker region aligns with the cylindrical shape of cilia, it suggests a tensegrity model, where the DMTs serve as compression elements, and the N-DRC represents tension elements. Thus, the tension of the N-DRC in situ serves a crucial role in maintaining the structural integrity of the axoneme. In this model, the DMTs bear compression forces as they form the core structure of cilia, while the N-DRC complex provides tension by connecting and regulating the movement of the DMTs. This tension-compression balance might be critical for maintaining the structural integrity and functionality of cilia.

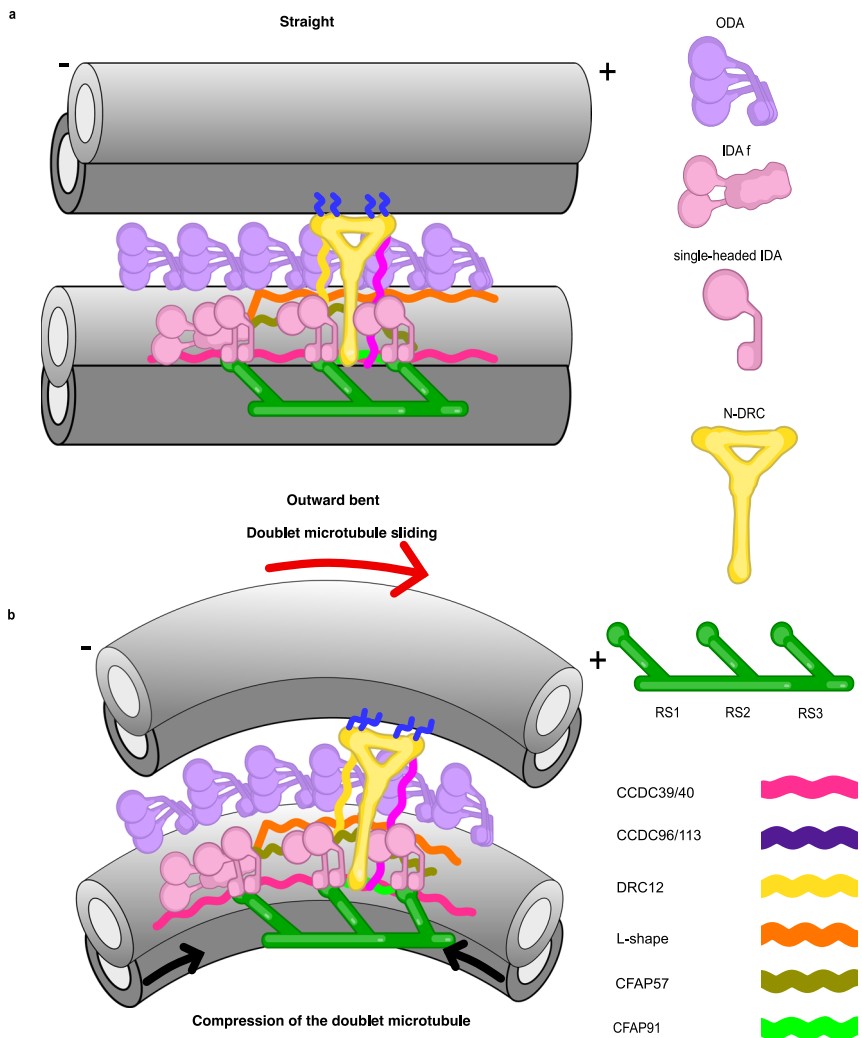

**Fig. 7 | Model for the conformational change of the N-DRC and the coiled-coil network during ciliary bending. a** Two neighboring DMTs are in straight conformations. **b** The N-DRC pulls due to DMT sliding and the coiled coils on the surface of the DMT due to bending, which can lead to dynein activation/inactivation. Signs (+) and (−) indicate the distal and proximal ends of the DMT.

## Mapping of pathogenic N-DRC human mutations allows the understanding of N-DRC assembly

In humans, mutations in the N-DRC scaffold components disrupt cilia motility and lead to primary ciliary dyskinesia[44,45]. To investigate the molecular basis of N-DRC-related ciliopathies, we mapped missense genetic mutations that cause ciliopathies on the human N-DRC model constructed based on the *T. thermophila* N-DRC model (Fig. 6a).

It has been reported that missense mutations (N276K, A391V) in DRC4 cause disorganization of DMT and change the swimming velocity. Our model suggests these mutations can lead to disorganization or disassembly of the DRC4 coiled-coil structure. A canonical coiled-coil is composed of heptad repeats, labeled *abcdefg*, where hydrophobic amino acids at positions *a* and *d* are conserved[46]. Asparagine is among the most abundant polar inclusions at positions *a* or *d*, providing conformational dynamics at the coiled-coil interface[47]. Our structural data suggest N276 is at position *a* in the coiled-coil heptad (Fig. 6b, c). Therefore, mutation of N276 to a positively charged lysine can disrupt or deform the DRC4 coiled-coil structure.

In summary, scaffold proteins (DRC1, DRC2, DRC4) play a vital role in assembling the N-DRC since any defect in these subunits can cause severe ciliopathies. These proteins construct the N-DRC core structure and interact with most proteins in the base plate and linker regions. Because of the pivotal role of scaffold proteins in N-DRC assembly, any defect in these proteins can cause loss of the N-DRC and disrupt the N-DRC's contact with the neighboring DMT. The contact of the N-DRC by electrostatic interactions with the neighboring DMT is essential for ciliary bending[36]. Previous studies on *C. reinhardtii* mutants of *drc7* and *drc11* show that the partially disrupted N-DRC still maintains contact with the neighboring DMT[21,36]. As a result, the bending motion is not completely disrupted if there are some contacts with neighboring DMTs. Therefore, pathogenic N-DRC mutations are primarily found in scaffold proteins.

## Discussion

In this study, we revealed the molecular architecture of the N-DRC of *T. thermophila* by an integrative modeling approach. Importantly, our data identified an additional N-DRC component, DRC12. We localized CFAP337 and CFAP57 and showed their interactions with the CCDC96/113 complex. The main scaffold of the N-DRC comprises coiled-coiled proteins, including DRC1/2 and DRC4A/4B complexes, which start from the base plate and span through the proximal lobe of the linker. These components play a pivotal role in stabilizing the N-DRC and contribute to mechano-regulatory signal transmission. The role of DRC1, DRC2, and DRC4 proteins as the core structural components of

the N-DRC agrees with earlier analyses of the N-DRC in *C. reinhardtii* (N-C-term tagging SNAP, BCCP)[41] and biochemical studies[18].

In contrast to the radial spokes and ODA, which are preassembled in the cytoplasm and transported into the cilium[48–50], a recent study revealed that the N-DRC is likely assembled step-by-step in the cilia since DRC2 and DRC4 are transported and assembled independently in the cilia[51]. In accordance with our model and mass spectrometry analysis of different knockout DRC mutants[18,24], scaffold proteins should assemble onto the DMT first before the other N-DRC subunits dock onto the core scaffold proteins. Loss of DRC3 in the *drc3* mutant strain led to the loss of part of the distal domain (based on our study, the distal domain lost in the *drc3* mutant represents DRC9/10 complex)[37]. We propose DRC3 binds to DRC4 first, leading to the assembly of DRC9/10 heterodimers. As a result, DRC4 plays a crucial role in assembling the N-DRC distal domain.

Our model implicates CCDC96 has crosstalk with the N-DRC. Therefore, the CCDC96/113 complex may function as a signal transfer bridge for the N-DRC that runs from radial spoke RS3 to DRC10 and vice versa. Regulatory signals can be sent to dynein e through the N-terminus of DRC9. In addition, DRC10 may send signals to dynein g through DRC3. Therefore, the DRC9/10 coiled-coil may be the important regulatory hub in the N-DRC that regulates dyneins g and e by receiving the signal from CCDC96/113.

Our model revealed that the N-DRC is connected to an extensive network of coiled-coil proteins on the DMT that span the 96-nm repeating unit. An essential aspect of coiled-coil protein function is mechanical stability against deformation forces[52]. Coiled-coil proteins can transmit mechanical loads such as the tails of myosin and kinesin motors or keratin bundles[52]. We hypothesized that the N-DRC conformation change during ciliary bending can be transmitted to the coiled-coil network by pulling force. To bend the axoneme, dyneins on one side should be inactive, while dyneins on the other side must be active[11]. During ciliary beating, this process should switch back and forth quickly, resulting in the high oscillating frequency of the cilium. When the cilium is in a straight conformation, the N-DRC is in its relaxed state (Fig. 7a). In this conformation, the N-DRC exerts no regulation on dynein arm activity on either side of the cilium. In the outward-facing curve of the cilium, the dyneins are in active states, leading to DMT sliding and N-DRC stretching due to the larger distance between neighboring DMTs (Fig. 7b). N-DRC stretching can pull on the highly elastic coiled-coil protein network. This can mechanically extend the length of coiled-coil proteins. As a result, the dynein arms can be regulated by the pulling of the coiled-coil proteins and switch to an inactive state. In active cilia, the N-DRC linker region is not well observed by cryo-ET, probably because of its flexibility[39]. As such, the extent of the deformation of the N-DRC linker region is still unclear and needs to be answered in the future.

To summarize, our study sheds light on the molecular mechanism of dynein arm regulation through the N-DRC and other mechano-regulatory complexes. Our results allow us to better understand the molecular etiology of ciliopathies and the mechanism of ciliary dysfunction.

## Methods

*T. thermophila* growth, cilium isolation by dibucaine treatment, cryo-EM sample preparation, and cryo-EM data acquisition were performed similarly to our previous work[26,27].

### *T. thermophila* culture

All *T. thermophila* strains in this study were grown in 1 L of SPP media in a shaker incubator at 30 °C and 120 rpm until the cell density reached an OD of ~0.7 at 700 nm[53].

### Cilia isolation

The *T. thermophila* culture was harvested by centrifugation at 700×*g* for 10 min at room temperature. The cell pellet was resuspended in fresh SPP at room temperature to a total volume of 24 ml. One milliliter of SPP containing 25 mg of dibucaine was added to the cell solution for 1 min for deciliation. Immediately, 75 ml of ice-cold SPP was added to the cell suspension. The solution was centrifuged at 2000×*g* for 10 min at 4 °C. The supernatant containing the cilia was then centrifuged at 17,000×*g* for 40 min at 4 °C. The cilia-containing pellets were washed with ice-cold cilia wash buffer (50 mM HEPES at pH 7.4, 3 mM MgSO₄, 0.1 mM EGTA, 1 mM DTT, 250 mM sucrose) and resuspended in 250 µl cilia wash buffer.

### Purification of DMTs

NP-40 was added to a final concentration of 1.5% and incubated on ice for 45 min. After centrifugation for 10 min at 7800×*g* and 4 °C, the intact axoneme with the membrane in the pellet was resuspended in cilia final buffer (50 mM HEPES at pH 7.4, 3 mM MgSO₄, 0.1 mM EGTA, 1 mM DTT, 0.5% trehalose). To induce sliding disintegration for intact DMT purification, ADP was added to a final concentration of 0.3 mM and incubated at room temperature for 10 min. Then, ATP was added to a final concentration of 1 mM and incubated at room temperature for another 10 min to maximize sliding disintegration.

### Cryo-EM sample preparation

Chloroform was applied overnight to C-Flat Holey thick carbon grids for cleaning (Electron Microscopy Services #CFT312-100). For single-particle analysis, chloroform-treated and negatively glow-discharged (10 mA, 15 s) grids were placed inside the Vitrobot Mk IV (Thermo Fisher) chamber, and 4 µl of the DMT sample at 2.2 mg/mL was applied to the grid. The sample was incubated on the grid at 23 °C for 15 s with 100% humidity, blotted with force 3 for 5 s, and then plunged frozen in liquid ethane.

### Cryo-EM data acquisition

Beam-shift single-particle cryo-EM data were collected by SerialEM[54] on a Titan Krios 300 kV FEG electron microscope (Thermo Fisher) equipped with a K3 Summit direct electron detector (Gatan, Inc.) and the BioQuantum energy filter (Gatan, Inc.) (Table 1). The beam-shift pattern used to collect movies is four movies per hole and four holes per movement. The final pixel size is 1.370 Å/pixel. A total dose of 45 or 70 electrons per Å² was radiated to each movie with 40 frames. The defocus range was between −1.0 and −3.0 µm at an interval of 0.25 µm.

### Cryo-EM image processing of the 96-nm repeat unit of the DMT

The movies were motion-corrected and dose-weighted using MosionCor2[55] in Relion 3.1.2[56], and the contrast transfer function parameters were determined using GCTF[57]. The filaments were picked semi-automatically. Due to the importance of the top views of filaments, these views were picked manually using e2helixboxer[58]. The side views of the DMT were picked automatically using topaz[59] in Cryosparc 3.1[60]. Four-times binned particles of 128 × 128 pixels were extracted in Relion 3.1.2. The pre-alignment of 8-nm repeat particles was done by the Iterative Helical Real Space Reconstruction script[61] in the SPIDER package[62]. In the next step, the particles were transferred to Frealign and refined[63].

After converting the alignment parameters to Relion 3.1.2, iterative per-particle-defocus refinement and Bayesian polishing of 8-nm repeat particles with an unbinned box size of 512 × 512 pixels were performed. The signal of the tubulin lattice was subtracted to highlight the signal of MIPs for the classification of 16-nm and 48-nm repeats. To get a 16-nm repeating unit, 3D classification was performed with two classes. The 16-nm repeat particles were subjected to 3D classification with three classes to obtain the 48-nm repeat. Refinement of the non-subtracted particles was performed to obtain a 48-nm structure. To obtain the 96-nm repeat of the DMT, K40R, MEC17, SB255, and CU428 data were merged since the N-DRC structures in the mutants were not different. Then, 3D classification

with two classes with a mask focusing on the outside of the DMT was performed on the 48-nm repeat structure. The refinement of the 96-nm class containing N-DRC resulted in a global resolution of 3.6–4.0 Å.

To improve the local resolution of the base plate part, we performed local refinement using a mask that covered the base plate region and approximately two protofilaments attached to the base plate. Then, the map was enhanced by DeepEmhancer[64].

### Cryo-EM data processing of the N-DRC

A processing pipeline is shown in Supplementary Fig. 1a. The N-DRC base plate was improved using focus refinement with one mask that included the base plate part and CCDC96/113 complex on the DMT in Relion 3.1.2. To obtain the linker region, the particles of the 96-nm repeat were transferred to Cryosparc 3.1 to perform tubulin lattice signal subtraction, IDA subtraction, and centering of the N-DRC structure in the box using the volume alignment tool in Cryosparc 3.1. The star files were converted to Relion 3.1.2 using the UCSF pyem package[65]. After consensus refinement of the entire N-DRC, the refined structure was subjected to 3D classification without alignment with five classes and a T value of 50. The best class with the highest number of particles and resolution was selected for further 3D refinement. Then, the N-DRC base plate was subtracted, and the box size was reduced to 150 pixels. The resulting particles were locally refined with three different masks covering different regions of the linker region. The final maps at 6.7 Å resolution were B-factor sharpened and enhanced by DeepEmhancer[64]. All maps and models were visualized using ChimeraX[66].

### Cryo-electron tomography sample preparation

To maintain the axoneme intactness for cryo-electron tomography, glutaraldehyde (at a final concentration of 0.15%) was used to cross-link the axonemes for 40 min on ice. The reaction was quenched using 1 M HEPES. The axoneme solution at 3.6 mg/mL concentration was mixed with 10 nm gold beads in a 1:1 ratio. 4 μL of the cross-linked axoneme mixture was deposited onto negatively glow-discharged (10 mA, 15 s) C-Flat Holey thick carbon grids (Electron Microscopy Services #CFT312-100) inside a Vitrobot Mk IV (Thermo Fisher) chamber. The sample was incubated for 15 s at 23 °C and 100% humidity, then blotted with a blot force of 0 for 8 s before plunge frozen in liquid ethane.

### Tilt series acquisition and tomogram reconstruction

Tilt series were acquired with the dose-symmetric scheme from −60 to 60 degrees in 3-degree increments, and a total dose is 160 e⁻ per Å². The defocus value for each tilt series ranged between −2.5 and −3.5 μm. A series of 10–13 frames were captured for each view. The pixel size for the tilt series is 2.12 Å. CTF estimation was done in WARP[67]. Fiducial detection and alignment were performed by IMOD[68]. Batch reconstruction of four-times binned tomograms was done using IMOD (Supplementary Fig. 6a). The cross-linked axonemes displayed similarity to non-cross-linked axonemes, devoid of noticeable artifacts.

### Subtomogram averaging

The DMT subtomograms were picked by IMOD[68]. The AxonemeAlign program was used to perform subtomogram averaging on the four-times binned 96 nm repeating unit of the axoneme (comprising 2608 subtomograms)[69]. The alignment parameters were then converted to Relion 4.0. Refinement of the 96 nm subtomogram averages was accomplished through the Relion 4.0 pipeline[70]. The resolution achieved for the 96-nm repeating unit axoneme was 18 Å. To improve the local resolution of the N-DRC, focused refinement was performed using a mask that covers the N-DRC, resulting in a 22.26 Å resolved map (Supplementary Fig. 6).

### Identification of N-DRC proteins in *T. thermophila*

To identify the DRC subunits of *T. thermophila*, the *C. reinhardtii* DRC subunits (obtained from previous studies and the UniProt and Phytozome databases[25,71]) were blasted against the local database of the *T. thermophila* axoneme proteome, which was built using proteins identified in mass spectrometry data from the *T. thermophila* axoneme[26,27,72,73]. The exponentially modified protein abundance index (emPAI) was considered another important factor in identifying DRC components (Supplementary Table 2). DRC4, DRC6 and DRC11 all have two paralogs, A and B. In the case of DRC6 and DRC11, in which one paralog is significantly more abundant, we used the more abundant paralogs DRC6A and DRC11A for modeling (Supplementary Tables 1 and 2).

The intramolecular and intermolecular cross-links from the N-DRC components came from in situ cross-linking data of *T. thermophila* cilia[28] (Supplementary Tables 3 and 4).

### AlphaFold2 structure prediction

Using the AlphaFold2 Google Colab notebook[74], we predicted all N-DRC components individually. To localize proteins and model heterodimer and homodimer proteins, we used AlphaFold2 Multimer[75] implemented in the Cosmic 2 computer cluster[76] and Google Colab notebook[74]. All models were predicted using the full sequence. The model was truncated and refined to fit into the cryo-EM map.

### Localization and modeling of N-DRC proteins

For base plate modeling, we used two different methods, including multiple sequence alignment and AlphaFold2 prediction. For the N-DRC base plate, multiple sequence alignment of DRC1, DRC2, CFAP91, and CFAP20 was performed against *C. reinhardtii* homologs. In the next step, the PDB structure of *C. reinhardtii* (PDB 7JU4) was modified into the sequence of *T. thermophila*. For DRC4A/4B, and CCDC96/113, we used AlphaFold2 prediction. The models were then fitted into the map using the UCSF ChimeraX function fitmap[66]. Next, the models were modeled using Coot[77] and real space refined in Phenix[78] (Table 2).

To localize DRC3, DRC9, DRC10, and DRC7, we performed AlphaFold2 Multimer protein complex prediction, which confirmed that DRC9 and DRC10 tightly bind together and form a heterodimer that interacts with DRC3 (Supplementary Fig. 2f). The structure of DRC3 was resolved as a bundle of coiled-coil proteins connected to a bundle of beta sheets with a loop in the middle of the linker region. This allowed us to fit the high-confidence AlphaFold2 structure into the density (Supplementary Fig. 2c). The DRC5 density was resolved as a leucine-rich density that interacts with the linker region of DRC1/2, consistent with a previous study[21]. The high-confidence AlphaFold2 multimer model of DRC1 and DRC5 confirms the interaction between these proteins (Supplementary Fig. 2c, f). In our cryo-EM map, the resolved density of the DRC9/10 coiled-coil has a distinct N-terminal fold, consistent with the molecular model predicted by the AlphaFold2 multimer. Therefore, we could fit the N-terminal regions and coiled-coil regions of the DRC9/10 coiled-coil entirely within our map (Supplementary Fig. 2e). The sufficient resolution of the linker region also allowed us to fit the AlphaFold2 multimer model of DRC1/2 and DRC4A/4B complexes into our map.

For the identification and localization of other proteins, only proteins with at least two kinds of localization evidence are discussed in the work. The evidence includes cross-links to well-localized proteins, biotinylated in BioID of known proteins, highly confident AlphaFold2 multimer prediction with known proteins, presence in a pull-down assay with known bait proteins, missing in knockout mutants of known proteins and good fitting in low-resolution maps.

The cross-linking data indicated DRC7 is cross-linked with DRC10 and DRC3. Hence, to localize DRC7, we performed AlphaFold2 multimer prediction of DRC7, DRC10, and DRC3 (Supplementary Fig. 2f). In

addition, the linker region of DRC1/2 was also localized using Alpha-Fold2 Multimer (Supplementary Fig. 2f).

To find DRC8 and DRC11 on the N-DRC structure, we used Assembline as integrative modeling software, AlphaFold2 Multimer, Hdock protein docking server, and cross-linking MS/MS data. A high-confidence AlphaFold2 Multimer prediction and the high score of the Hdock docking server and Assembline[79–83] enabled us to fit the DRC8/11 complex in the N-DRC proximal lobe (Supplementary Fig. 2f). Using the UCSF ChimeraX function fitmap and Assembline Global optimization function, two copies of DRC8 and one copy of DRC11 were localized in our map. The fit libraries were created with 100,000 searches in the cryo-EM map using the fitmap tool of the UCSF Chimera[84] with the requirement of at least 60% of the input structure being covered by the EM map envelope defined at a low-density threshold. This resulted in 500–2800 alternative fitting in the cryo-EM map. Then, the resulting fit libraries of the DRC8 and DRC11 structures were used as input for the simultaneous fitting of DRC8 and DRC11 into the cryo-EM map. The fitting was performed using simulated annealing Monte Carlo optimization that generates alternative configurations of the fits pre-calculated as above.

DRC4 in *T. thermophila* has two paralogs compared with *C. reinhardtii*. Our cryo-EM map shows two coiled-coil densities for the DRC4 paralogs, in which one coil has a longer loop DRC4A (Q23YW7) with a particular fold and the other coil has a shorter loop DRC4B (I7LT80) (Supplementary Fig. 5f). To identify how these two paralogs interact with each other and to distinguish which density on our map belongs to these proteins, we performed AlphaFold2 multimer prediction. Interestingly, AlphaFold2 Multimer predicted the dimer with high accuracy, allowing us to localize the proteins (Supplementary Fig. 5d).

### Visualization of axonemal dyneins and DMT interaction
To visualize the dynein and axonemal dynein interaction, the isolated *T. thermophila* outer dynein arm model (PDB 7K58) was fitted into the 96-nm repeat subtomogram average (EMD-29667). For IDA, the head and stalk domains of *T. thermophila* dynein heavy chain DYH10 (UniProtID I7LTP7) were modeled using the Phyre2 server[85]. This model was then used to fit into all eight different inner arm dynein positions (a, b, c, d, e, g, fα, and fβ). The fitting is based on the head and orientation of the stalk densities in the subtomogram average.

### RNA expressions data analysis
Normalized transcript per million (nTPM) RNA expression levels of DRCs in different human tissues were obtained from The Human Protein Atlas available at www.proteinatlas.org. (Retrieved on December 20, 2022). Statistical correlation tests and virtualization were done using Prism v.6 software (GraphPad Software, La Jolla, CA, USA). nTPM values are shown in Supplementary Table 6.

### Genome modifications
To engineer *T. thermophila* cells expressing C-terminally -3HA or -HA-BirA*-tagged DRC fusion proteins under the control of the respective native promoter, the FAP44 open reading frame and 3′ UTR fragments were removed from FAP44-3HA and FAP44-HA-BirA* plasmids[86] using MluI and BamHI or PstI and XhoI sites, respectively, and replaced by approximately 1 kb fragments of the C-terminal part of the open reading frame without a stop codon and 3′UTR amplified using wild-type genomic DNA as a template and Phusion Hot Start II high-fidelity DNA polymerase (Cat# F549L, Thermo Fisher Scientific, Vilnius Lithuania) and primers listed in Supplementary Table 7.

To engineer *T. thermophila* cells over-expressing C-terminally GFP- or HA-tagged proteins under the control of a cadmium-inducible MTT1 promoter, the entire open reading frame without the stop codon was amplified with the addition of the MluI (Cat# E2288, Eurx, Gdansk, Poland) and BamHI (Cat# E2050, Eurx, Gdansk, Poland) restriction sites at the 5′ and 3′ ends, respectively, using Phusion Hot Start II high-

fidelity DNA polymerase and the primers listed in Supplementary Table 7. The PCR fragments were cloned using MluI and BamHI restriction enzymes into modified versions of the pBlueScript II KS(+)_MTT1-GFP or pBlueScript II KS(+)_MTT1-HA plasmids[87], enabling the integration of the transgene into the *BTU1* locus and selection of transformed *T. thermophila* cells based on paromomycin resistance (neo2 cassette)[88]. *T. thermophila* cells were transformed, and transgenes were assorted as described[12,30]. Constructs used to engineer *T. thermophila* DRC mutants were obtained in the laboratory by cloning fragments of the genomic DNA amplified by PCR with the addition of appropriate restriction enzymes. Proper cloning was confirmed by the restriction analyses and DNA fragment sequencing. Constructs were introduced to the *T. thermophila* cells by biolistic transformation, and positive transformants were selected based on the resistance to paromomycin. The expression of the proteins was confirmed by immunofluorescence and western blotting to compare the molecular weight of the expressed proteins with the theoretical molecular weight. Three to six independent clones were analyzed for each DRC mutant.

### BioID assay
For the BioID assay, cells were grown to a density of $2 \times 10^5$ cells/ml, starved for 14–18 h in 10 mM Tris-HCl buffer, pH 7.5, and incubated in the same buffer supplied with 50 μM biotin for 4 h at 30 °C. Next, the cells were spun down and deciliated, and the collected cilia were resuspended in 0.5 ml of axoneme stabilization buffer (20 mM potassium acetate, 5 mM MgSO$_4$, 20 mM HEPES, pH 7.5, 0.5 mM EDTA with protease inhibitors (Complete Ultra EDTA-free; Roche, Indianapolis, IN)). After incubation of cilia for 5 min on ice in the same buffer supplied with 0.2% NP-40, the axonemes were pelleted by centrifugation at 21,100×*g* for 10 min at 4 °C and lysed for 1 h (0.4% SDS, 50 mM Tris-HCl, pH 7.4, 500 mM NaCl, 1 mM DTT with protease inhibitors) at RT. After centrifugation (8000×*g* at 4 °C), the supernatant was collected and diluted with three volumes of 50 mM Tris-HCl buffer, pH 7.4. Collected proteins were incubated overnight with 100 μl of Dynabeads M-280 Streptavidin magnetic beads (Cat# 11205D Thermo Fisher Scientific, Waltham, MA, USA) at 4 °C. After washing with buffer containing 15 mM Tris-HCl, pH 7.4, 150 mM NaCl, 0.1% SDS, and 0.3 mM DTT at 4 °C, the biotinylated, resin-bound proteins were analyzed by mass spectrometry (Laboratory of Mass Spectrometry, Institute of Biochemistry and Biophysics, PAS, Warsaw, Poland) and by western blotting using Pierce High Sensitivity-streptavidin-HRP (Cat# 21130 Thermo Scientific, Rockford, IL, USA) diluted 1:40,000 in 3% BSA/TBST.

### Pull-down analyses
For the pull-down assay, cells carrying a transgene enabling over-expression of *T. thermophila* proteins were grown overnight to the mid-log phase in SPP medium and diluted to a density of $2–2.5 \times 10^5$ cells/ml, and overexpression was induced by the addition of cadmium chloride to a final concentration of 2.5 μg/mL. After 2–3 h, cells were collected, washed in 10 mM Tris-HCl buffer, pH 7.5, and solubilized in two-fold concentrated modified RIPA buffer (50 mM Tris-HCl, pH 7.5, 300 mM NaCl, 2% NP-40, 2% sodium deoxycholate, 10% glycerol with protease inhibitors).

After spinning down at 100,000×*g* for 30 min at 4 °C, the supernatant obtained from cells expressing GFP-tagged proteins was diluted 1:4 with dilution buffer (25 mM Tris-HCl, pH 7.5, 150 mM NaCl, 5% glycerol with protease inhibitors), and ~250 μg of protein was incubated with GFP-Trap® Magnetic Agarose beads (Cat# gtma, Chromo-Tek, Proteintech, Planegg-Martinsried, Germany) for 1–1.5 h. Next, beads were washed (5 × 5 min) with Wash buffer (25 mM Tris-HCl, pH 7.5, 150 mM NaCl, 0.2% NP-40.5% glycerol with protease inhibitors) and incubated with ~250 μg of supernatant containing HA-tagged proteins prepared as described above for GFP fusion proteins. After 1–1.5 h of incubation at room temperature, followed by washing with wash

 

buffer, the bead-bound proteins were analyzed by western blotting using rabbit anti-GFP (Cat# Ab6556, Abcam, Cambridge, UK) and mouse anti-HA (Cat# 901503, BioLegend, San Diego, CA, USA) antibodies.

## Western blot

In case of the pull-down experiments entire resin with bound proteins obtained after final washing step was mixed with 40 μl of 2.5 times concentrated Laemmli buffer, and samples were denatured for 5 min at 95 °C. Samples (20 μl) were run onto 10% polyacrylamide gel (SDS-PAGE, gel's thickness = 1.0 mm). The electrophoresis was performed using an electrophoresis buffer (25 mM Tris-HCl, 192 mM glycine, 0,1% SDS, pH 8.3) in a Mini-Protean II apparatus (Bio-Rad, USA), at a voltage of 120–150 V for ~90 min. The PageRuler Prestained protein Ladder, 10–180 kDa (Thermo Scientific, Catalog No. 26617) was used as a marker to determine the relative molecular weight of the test proteins.

The separated proteins were transferred to a nitrocellulose membrane (Bio-Rad, USA) using a wet transfer method in a Mini Trans-Blot apparatus (Bio-Rad, USA) and transfer buffer (25 mM Tris-HCl, 192 mM glycine, 20% methanol, pH 8.3). Transfer was performed at a constant current of 170 mA for 70 min on ice. After transfer, the membrane was blocked for 1 h with 5% skimmed milk in TBST (10 mM Tris-HCl, 150 mM NaCl, pH 8 supplemented with 0.1% Tween-20) and next incubated overnight at 4 °C with the 1:2000 dilution of mouse anti-HA antibody (Cat# 901503, BioLegend, San Diego, CA, USA) or 1:60,000 dilution of purified polyclonal rabbit anti-GFP antibody (Cat# Ab6556, Abcam, Cambridge, UK) diluted in 5% skimmed milk in TBST. After washing for 4 × 10 min in TBST, the membrane was incubated with respective secondary antibodies, either 1:10,000 dilution of HRP-conjugated anti-mouse IgG (Cat# 115-035-146, Jackson ImmunoResearch Laboratories, UK) or 1:20,000 dilution of HRP-conjugated goat anti-rabbit IgG (Cat# 401315, Millipore, Darmstadt, Germany) in 5% skimmed milk for 1 h at RT. After washing in TBST (3 × 10 min) and once in TBS (10 min), proteins were detected using a Westar ηC Ultra 2.0 kit (Cat# XLS075, Cyanagen, Bologna, Italy). Membranes were visualized using G:BOX gel documentation system (Syngene, Cambridge, UK).

For BioID, ~1 mg of the axonemal proteins were incubated with 100 μl of streptavidin-coupled beads. After washing, proteins from 10% of the beads were analyzed by western blot. Proteins were separated on 10% gel (SDS-PAGE) and transferred onto nitrocellulose as described above. Blots were blocked with 3% BSA in PBS and detected using Pierce High Sensitivity-streptavidin-HRP (Cat# 21130, Thermo Scientific, Rockford, IL, USA) diluted as 1:40,000 in 3% BSA/TBST.

## Reporting summary

Further information on research design is available in the Nature Portfolio Reporting Summary linked to this article.

## Data availability

The data that support this study are available from the corresponding authors upon request. Maps generated have been deposited in the Electron Microscopy Data Bank (EMDB) under accession codes EMD-41284 (96-nm repeat of the DMT), EMD-41189 (combined N-DRC base plate), EMD-41251 (N-DRC linker region), EMD-41270 (subtomogram average of the N-DRC). Atomic models have been deposited in the Protein Data Bank (PDB) under accession codes 8TEK (N-DRC base plate), 8TH8 (N-DRC linker region), and 8TID (entire N-DRC and associated proteins). Previously published structures utilized are available from the PDB and EMDB: 7K58 (Outer-arm dyneins of *T. thermophila*), EMD-12121 (96 nm repeat of the *T. thermophila* CCDC96-deletion DMT), EMD-20338 (96 nm repeat of the *C. reinhardtii* DMT), EMD-5950 (96 nm repeat of the human DMT) and Dryad p2ngf1vqv (Mass spectrometry of DMT from *T. thermophila* WT and mutants). The original gel and blot, BioID and cross-linking mass spectrometry of WT generated in this study in Fig. 1e, Supplementary Fig. 4ac–e, Supplementary Fig. 5a, b are provided in the Supplementary Information and Source Data file. Source data are provided with this paper.

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

## Acknowledgements

We thank Drs. Kelly Sears, Mike Strauss, Kaustuv Basu (Facility for Electron Microscopy Research at McGill University) for helping with data collection, Dr. Jan Kosinski for helping to set up Assembline, and Drs. Maureen Wirschell and Muneyoshi Ichikawa for critically reading the manuscript. K.H.B. is supported by grants from the Canadian Institutes of Health Research (PJT-156354) and the Natural Sciences and Engineering Research Council of Canada (RGPIN-2022-04774). EMM acknowledges support from the Welch Foundation (F1515) and U.S. National Institutes of Health (R01 HD085901). D.W. is supported by a grant from the National Science Centre, Poland (OPUS21 2021/41/B/NZ3/03612).

## Author contributions

Conceptualization: K.H.B. and D.W. Performed structural biology experiments: A.G., C.S.B., S.K.Y., and T.L. Built and refined structural models: A.G. Performed biochemistry experiments: S.M., C.L.M., O.P., and M.J., M.V.P. Performed mass spectrometry experiments: C.L.M., O.P., and M.J. Formal analysis: A.G., S.M., and C.L.M. Statistics tests: B.N. Investigation: A.G., S.M., C.L.M., B.N., C.S.B., S.K.Y., T.L., O.P., M.J., M.V.P., E.M.M., D.W., and K.H.B. Resources: D.W., K.H.B., and E.M.M. Writing—original draft: K.H.B., D.W., and A.G. Writing—review and editing: K.H.B., D.W., A.G., and E.M.M.

## Competing interests

The authors declare no competing interests.
