## [Peer Review File · Nature Communications]

Integrated modeling of the Nexin-dynein regulatory complex reveals its regulatory mechanismReviewers' Comments:

Reviewer #1:

Remarks to the Author:

General assessment

In this manuscript, Ghanaeian et al. revealed the whole architecture of *Tetrahymena* N-DRC (nexin-dynein regulatory complex) by cryo-EM analysis. N-DRC is the Y-shaped protrusion attached to the doublet microtubules (DMT) in the axoneme and bridges neighboring DMTs to restrict the sliding motions between DMTs. Due to its flexibility, the linker region structure of N-DRC was not well resolved ($\sim 5\text{-}10 \text{ \AA}$) compared to the rigid base plate region (3.6 \AA). However, the author's integrative modeling approach fitted the models of all known N-DRC components into the cryo-EM map and identified an additional N-DRC component, DRC12 (CCDC153). The generated model agrees with the biochemical cross-linking data and the previous *Chlamydomonas* N-DRC studies, providing a comprehensive summary of the N-DRC structure.

Moreover, the authors searched for any linkage from the N-DRC to other DMT components, using the generated N-DRC model. The N-DRC was connected to a network of coiled-coil proteins on the DMT, which interacts with OADs, IADs, and RSs. Direct interactions of N-DRC components with OAD (outer arm dynein), IADe (inner arm dynein e), and IADg were also found. The diagram in Figure 5C summarizes the connections of the DRC components, coiled-coils, OADs, IADs, and RSs. These data give us critical information to understand the mechanical signal transduction from the N-DRC to the axonemal dyneins through the DMT's coiled-coil network.

Although a recent study already reported the model of a whole 96nm-repeat DMT structure containing N-DRC (*Chlamydomonas* and human; Walton et al., 2023), this manuscript focused on the N-DRC to comprehend the interactions of N-DRC with other DMT components, the evolutionary divergence of N-DRC structures, and the pathogenic basis of N-DRC-related ciliopathies. This manuscript is well-written, and the illustrations are clear. This work is worthy of publication in *Nature Communications*.

Main points

1)

Identification of DRC12 (CCDC153) is one of the key progresses of this work. For the clarification and validation of the DRC12 identification process, Figure S4A (the cross-linking positions between DRC1 and DRC12) should be shown in the main figures.

2)

To support the assignment of CCDC153 as a DRC protein, the authors also show the correlation of the RNA expression of CCDC153 with DRC1 and DRC2 (Figure 1F).

2)-1

The source of the RNA expression data should be clarified in the Materials and Methods section.

2)-2

The expression correlation of CCDC153 with other DRC proteins does not sufficiently support the assignment of CCDC153 as a DRC protein. For example, CCDC153 does not show the significant expression correlation with DRC4, DRC5, DRC6, DRC7, DRC10, and DRC11 (Figure S4E). Further explanation is required to understand the meaning of these data.

3)

Any FSC (Fourier shell correlation) plot is not provided in this manuscript. The FSC plots of each local refinement and the local resolution maps are required to validate the structural analysis of N-DRC.

4)

The connection of N-DRC to the adjacent DMT is essential for axonemal construction and bending motion. The authors found two connections between N-DRC and adjacent DMT in the subtomogram-averaged map (Figure 4A). However, the relationships of these connection sites to the N-DRC model are not clarified. Fitting the N-DRC model to the subtomogram averaged map could provide a discussion about the molecular basis of these connections. Moreover, comparing the generated N-DRC model with the subtomogram averaged map is important to consider the artifact of DMT isolation, since the single particle analysis of N-DRC used the isolated DMTs, whose N-DRC connections to the adjacent DMT were disrupted.

Minor points

>48 It is

>49 composed of a bundle of nine doublet microtubules (DMTs) composed of A- and B-tubules

>50 surrounded by two singlet microtubules known as the central pair (Fig. 1A).

"surrounded by" => "surrounding"

>70 Chlamydomonas

=> "Chlamydomonas"

>132 (Fig. S4B)

=> "(Fig. S4E)"

>217 trypanosomes

=> "Trypanosomes"

>363 The red line represents the N-DRC components.

There is no "red line" in Figure 5C.

>419 We localized CFAP337 and CFAP5 and demonstrated their interactions

>420 with the CCDC93/113 complex.

"CFAP5" => "CFAP57"

"CCDC93/113" => "CCDC96/113"

Reviewer #2:

Remarks to the Author:

This manuscript reports the cryo-EM structure of the nexin-dynein regulatory complex (N-DRC) in *Tetrahymena* cilia. In addition to the structural information, it includes crossing linking experiments and immunoprecipitation experiments to verify some of the interactions predicted from the structure. This is a major strength of this manuscript as these biochemical experiments are not seen in many of the published papers on the structure of cilia obtained using cryo-EM.

Comments

Motile cilia have over 800 proteins based on proteomics. The field has started to report that the structure of the motile cilia is solved. This is far from true. The structures reported have included fewer than two hundred proteins. For structures that repeat every 96 nm, the resolution that is obtained is lower than for proteins that repeat at 8 or 16 nm. Many of the structures reported in this manuscript have been reported previously. This manuscript actually tests some of the interactions. They could have taken more advantage the information that comes from previous mutant studies. The use of AlphaFold is important but it is also important to validate the predictions using biochemical

means.

I have a few questions and concerns.

The information present in Figure 1D shows structures but in Line 523 it is stated that a resolution of 5.7Å was obtained. How do the authors obtain this kind of detail with a resolution of 5.7 Angstroms? I would like to know what fraction of the amino acid side chains can be identified. In other words what fractions of the residues for each of the 12 N-DRC proteins could be identified. How much of Figure 1D would be grey if those amino acids whose side chains are not identified are colored grey?

In the abstract, the authors state that CCDC96 and CDC113 are in close contact with the N-DRC as if this were a new finding. In Bazan et al., 2021, it is stated these proteins form a connection with the N-DRC. Thus, this point does not seem to be new and does not warrant putting it in the abstract.

The last line of the abstract states that the N-DRC is associated with a network of coiled-coil proteins that may regulate the N-DRC. How does the information in this manuscript support this point? It seems to be a hypothesis that is never tested.

Lines 69-71. The authors state that loss of the DRC proteins rescues paralysis of mutants lacking the radial spokes and central apparatus. Based on the references they cite, these drc mutants suppress mutants that have lost the radial spokes. None of the references report suppression of the central apparatus mutants. Not all drc mutant suppress. Line 69. DRC should be in italics as the authors are discussing a gene and not the protein. This suggests that the authors are not familiar with the literature on this structure in cilia.

In Figure 6, the authors model human variants. Nonsense mutations are usually associated with a reduction in the steady-state level of cytoplasmic mRNA. This mechanism of nonsense mediated decay (NMD) is responsible for the degradation of mRNAs that contain a premature termination codon at a position at least 50 nucleotides upstream of an exon-exon boundary but it is not universal. Since most of the human variants do not have mRNA or protein data, modeling of nonsense variants does not make sense to learn about biological function. It is likely that the premature termination codons result in null alleles and not truncated proteins. Unless there is mRNA or protein data to support the variants in Figure 6A, I would suggest deleting this part of the figure. The missense variants in Figure 6B in DRC2 and DRC4 could be useful. It is possible that the variants destabilize the protein. Is there antibody or proteomics to show that the protein is present in the cilia or in the cell body in these variants? Does the patient phenotypes come from the loss of the protein or from a change in the interaction? Again, it is important to know if the protein is present or just destabilized and lost.

Figure 6B is so small that it does not provide any useful information. If Panel A is deleted then, each of the missense variants could be made bigger and the consequence of the missense variant shown by comparing the wild-type and variant proteins using AlphaFold.

Lines 233-280:

The interaction of DRC3 with dynein g was reported in 2015. Interactions with dynein e have previously been reported. Although there is increased resolution, the authors should acknowledge that other methods have already found many of these interactions. Discussions of crosstalk and signaling are better for a discussion than the in the results. There are no experimental results about this proposed signaling.

Figure 3 discusses the protein FAP337. Crosslinking shows interaction with CCDC96. FAP337 is a protein that is reduced in the Chlamydomonas mutant fap57 while CCDC96 is not missing in the fap57 mutant. Do the authors see evidence of FAP57 or FBB7 (a paralog) in their structures or in their cross-linking experiments? Dynein d and g are reduced in the fap57 mutant as well. If there is a network, these results suggest the network is not reciprocal.

Minor points

What is a linker part? This term is used repeatedly and I do not know what it means. It is on line 81, 103, 110, 164, 180, 192 and many more. If it is to be used, it should be defined.

In Figure 1C, the grey is not identified. Please include an explanation. The text states that 85% of the analyzed density is explained; is this the grey regions?. Are these parts that could not be identified with respect to the tomography images of the N-DRC?

The colors used for AlphaFold2_CrossLink and Coot are hard to tell apart. I am not sure if there are two different colors in the cleft or just one.

Line 45: Is there any evidence that motile cilia affect the development and maintenance of the spinal cord in humans? Reiter and Leroux suggest a low frequency of headaches and hydrocephaly.

Line 50: central pair. The field is moving towards calling the two singlet microtubules the central apparatus since it is much more than the pair of microtubules. It is likely to have over 70 proteins and many associated structures. Please consider using central apparatus and not central pair.

Line 58: The number of dynein is organism specific. There are two outer dyneins in humans but three in *Chlamydomonas* and *Tetrahymena*. There are six inner dynein arms in *Chlamydomonas*, there are reported to be

Line 134-136: It is interesting that there are paralogs of DRC4. How different are DRC4A and DRC4B? Are the difference in regions that do not interact in the structure?

Line 176: The original study of FAP91 was by Dymek, Heuser, Nicastro and Smith (2011) and this reference should be cited.

Line 268: Loss of proteins in a mutant suggest that the assembly requires the mutant protein but does not show interaction.

DRC12 is reported to be an N-DRC protein. *Chlamydomonas* has no ortholog and proteomics of isolated airway cilia do not show its presence. Is there any data to support this protein in cilia or flagella?

We would like to thank all the reviewers for their constructive comments. We have revised the papers to address the concerns of the reviewers. In summary, here are what we included in the revised version.

- We did further data analysis, improved the clarity in our writing and the figures according to the reviewers' suggestion and our own judgement.
- Our collaborators have improved their crosslinks search and presented an improved crosslink list (<https://www.biorxiv.org/content/10.1101/2023.07.09.548259v1>). We incorporated this information into the manuscript and found some more crosslinks within the DRC for modelling.
- We added a supplementary table S1, summarizing the rationale for the assignments of proteins in this study.
- One new insight that we added to the manuscript is the N-DRC structure tilts 6 degrees between our cryo-EM model and the N-DRC in the intact cilium. This means the N-DRC inside the cilium is already has stressed built in the structure compared to the relaxed purified doublet microtubule.
- To increase the transparency of our research work, we deposited the cryo-EM data into the EMDB and the model into PDB.

The newly added texts are highlighted in the manuscript for easy viewing.

REVIEWER COMMENTS

Reviewer #1 (Remarks to the Author):

General assessment

In this manuscript, Ghanaeian et al. revealed the whole architecture of Tetrahymena N-DRC (nexin-dynein regulatory complex) by cryo-EM analysis. N-DRC is the Y-shaped protrusion attached to the doublet microtubules (DMT) in the axoneme and bridges neighboring DMTs to restrict the sliding motions between DMTs. Due to its flexibility, the linker region structure of N-DRC was not well resolved (~5-10 Å) compared to the rigid base plate region (3.6 Å). However, the author's integrative modeling approach fitted the models of all known N-DRC components into the cryo-EM map and identified an additional N-DRC component, DRC12 (CCDC153). The generated model agrees with the biochemical cross-linking data and the previous Chlamydomonas N-DRC studies, providing a comprehensive summary of the N-DRC structure.

Moreover, the authors searched for any linkage from the N-DRC to other DMT components, using the generated N-DRC model. The N-DRC was connected to a network of coiled-coil proteins on the DMT, which interacts with OADs, IADs, and RSs. Direct interactions of N-DRC components with OAD (outer arm dynein), IADe (inner arm dynein e), and IADg were also found. The diagram in Figure 5C summarizes the connections of the DRC components,

coiled-coils, OADs, IADs, and RSs. These data give us critical information to understand the mechanical signal transduction from the N-DRC to the axonemal dyneins through the DMT's coiled-coil network.

Although a recent study already reported the model of a whole 96nm-repeat DMT structure containing N-DRC (*Chlamydomonas* and human; Walton et al., 2023), this manuscript focused on the N-DRC to comprehend the interactions of N-DRC with other DMT components, the evolutionary divergence of N-DRC structures, and the pathogenic basis of N-DRC-related ciliopathies. This manuscript is well-written, and the illustrations are clear. This work is worthy of publication in Nature Communications.

We thank the reviewer for acknowledging the significance of our research. We did our analysis independently from the 96-nm repeat DMT from *Chlamydomonas* and human (Walton et al., 2023) and used other techniques to help us confidently assign the proteins. As a result, we were able to identify DRC12 and CFAP337 as protein interacting with CCDC96/113 and CFAP57. Having an independent study on *Tetrahymena* allows us to see the conserved and non-conserved features in the N-DRC, which will be essential to understand N-DRC functions.

Main points

1)

Identification of DRC12 (CCDC153) is one of the keys progresses of this work. For the clarification and validation of the DRC12 identification process, Figure S4A (the cross-linking positions between DRC1 and DRC12) should be shown in the main figures.

The crosslink validation of DRC12 is now added to Figure 1F. In addition, we also added Table S1 to clarify the rationale for assignment of the protein in this study.

2)

To support the assignment of CCDC153 as a DRC protein, the authors also show the correlation of the RNA expression of CCDC153 with DRC1 and DRC2 (Figure 1F).

2)-1

The source of the RNA expression data should be clarified in the Materials and Methods section.

The statement has been revised and the origin of the data has been referred.

Normalized transcript per million (nTPM) values for each DRC in different human tissues were obtained from The Human Protein Atlas available at www.proteinatlas.org. (Data was retrieved on December 20, 2022).

2)-2

The expression correlation of CCDC153 with other DRC proteins does not sufficiently support the assignment of CCDC153 as a DRC protein. For example, CCDC153 does not show the significant expression correlation with DRC4, DRC5, DRC6, DRC7, DRC10, and DRC11 (Figure S4E). Further explanation is required to understand the meaning of these data.

It is true that the expression correlation alone is not sufficiently support the assignment of CCDC153 as a DRC protein. Now, with table S1, we listed multiple evidence of the assignment of CCDC153 as a DRC protein and the expression correlation is one of them since a valid indicator of physical or functional interactions between two or more proteins is co-expression in a context-dependent manner such as specific tissue types.

1. Our map showed unidentified coiled-coil density which interacts with DRC1/2 and DRC6.
2. DRC12 cross-link with DRC1 and DRC6C.
3. AlphaFold model of DRC12 match with the coiled-coil density in our map.
4. The positive charge of flexible loops in top part of the linker domain of DRC12 is consistent with the proposed electrostatic interaction between the NDRC and polyglutamylated tubulin[9].
5. Bio-ID result showed that DRC12 biotinylated when DRC1-HA-BirA* and DRC2-HA-BirA* tagged as baits.
6. The RNA expression of DRC12 from different human tissues correlate well with that of DRC1 and DRC2.

.

We updated the text to cite this supplementary table.

3)

Any FSC (Fourier shell correlation) plot is not provided in this manuscript. The FSC plots of each local refinement and the local resolution maps are required to validate the structural analysis of N-DRC.

FSC plots now added to Figure S1.

4)

The connection of N-DRC to the adjacent DMT is essential for axonemal construction and bending motion. The authors found two connections between N-DRC and adjacent DMT in the subtomogram-averaged map (Figure 4A). However, the relationships of these connection sites to the N-DRC model are not clarified. Fitting the N-DRC model to the subtomogram averaged map could provide a discussion about the molecular basis of these connections. Moreover, comparing the generated N-DRC model with the subtomogram averaged map is important to consider the artifact of DMT isolation, since the single particle analysis of N-DRC used the isolated DMTs, whose N-DRC connections to the adjacent DMT were disrupted.

We added a panel to fig S7 addressing this point.

We also added this to the text to highlight:

“Fitting the pseudo-atomic model of N-DRC in subtomogram average showed that the density which connect to PF B8 corresponded to DRC6 (Fig. S7C). Previous studies showed that *Supp4* mutation, which lacks DRC5 and DRC6 rescues the paralysis caused by defect in radial spoke [37]. Therefore, our structure suggests that connection of DRC6 to the adjacent DMT can act as a brake for dynein arms since lack of this protein can rescue the movement of cilia.”

Minor points

>48 It is

>49 composed of a bundle of nine doublet microtubules (DMTs) composed of A- and B-tubules

>50 surrounded by two singlet microtubules known as the central pair (Fig. 1A).

“surrounded by” => “surrounding”

>70 Chlamydomonas

=> “Chlamydomonas”

>132 (Fig. S4B)

=> “(Fig. S4E)”

>217 trypanosomes

=> “Trypanosomes”

>363 The red line represents the N-DRC components.

There is no “red line” in Figure 5C.

>419 We localized CFAP337 and CFAP5 and demonstrated their interactions

>420 with the CCDC93/113 complex.

“CFAP5” => “CFAP57”

“CCDC93/113” => “CCDC96/113”

We fixed them.

Reviewer #2 (Remarks to the Author):

This manuscript reports the cryo-EM structure of the nexin-dynein regulatory complex (N-DRC) in Tetrahymena cilia. In addition to the structural information, it includes crossing linking experiments and immunoprecipitation experiments to verify some of the interactions predicted from the structure. This is a major strength of this manuscript as these biochemical

experiments are not seen in many of the published papers on the structure of cilia obtained using cryo-EM.

Comments

Motile cilia have over 800 proteins based on proteomics. The field has started to report that the structure of the motile cilia is solved. This is far from true. The structures reported have included fewer than two hundred proteins. For structures that repeat every 96 nm, the resolution that is obtained is lower than for proteins that repeat at 8 or 16 nm. Many of the structures reported in this manuscript have been reported previously. This manuscript actually tests some of the interactions. They could have taken more advantage the information that comes from previous mutant studies. The use of AlphaFold is important but it is also important to validate the predictions using biochemical means.

We appreciated the positive point raised by reviewer #2 regarding the use of biochemical experiments in our manuscript. We also agreed that the structure of the motile cilia is still far from solved, especially only with cryo-EM. Apart not stably bound to the axoneme or binding to the axoneme without periodicity, there are certainly radial asymmetry or longitudinal asymmetry along the cilium as shown by many studies using fluorescence proteins or cryo-ET. Those asymmetrical binding proteins are not as easy to solve, identify and localize due to the lack of resolutions. As a matter of fact, we believe that our manuscript with the usage of various techniques can serve as a base structure and a novel method for the identification of asymmetrically localized proteins in the future. Using crosslinking mass spectrometry, BioID and pulldown assay, we showed that it is possible to identify new proteins and validate them. With further technical improvement in crosslinking mass spectrometry and BioID, it is possible to identify more proteins do not present in all the repeating units of the cilia. Already in this revised manuscript with the improvement of the crosslinking, we also listed some new crosslinks from the DRC to proteins in the N-DRC.

I have a few questions and concerns.

The information present in Figure 1D shows structures but in Line 523 it is stated that a resolution of 5.7Å was obtained. How do the authors obtain this kind of detail with a resolution of 5.7 Angstroms? I would like to know what fraction of the amino acid side chains can be identified. In other words what fractions of the residues for each of the 12 N-DRC proteins could be identified. How much of Figure 1D would be grey if those amino acids whose side chains are not identified are colored grey?

We updated this information precisely Figure S2A and Table S1. Basically, we can see the side chain of part of proteins in the baseplate. Any protein outside the baseplate is modelled by docking AlphaFold predicted structures only. Therefore, the side chain information there is not accurate.

In the abstract, the authors state that CCDC96 and CDC113 are in close contact with the N-DRC as if this were a new finding. In Bazan et al., 2021, it is stated these proteins form a connection with the N-DRC. Thus, this point does not seem to be new and does not warrant putting it in the abstract.

Bazan et al., 2021 indicates the density connection from CCDC96/113 to the N-DRC, it is in fact not accurate due to the low-resolution structure. There is not direct density link between the CCDC96/113 to N-DRC as indicated in Bazan et al. In our work, we identified the N-terminal globular region of CCDC96 contribute to the density of the distal region of the N-DRC. Between this region and the modelled part of the CCDC96, the protein is disordered and therefore no density observed. Since we could localize the N-terminus of the CCDC96 until the disorder region, we could rationally predict that the N-terminal globular region of CCDC96 is in the distal density of the N-DRC. Using the WT and CCDC96-coDel mutant cryo-EM maps from Bazan et al. help us to clarify the location of the globular N-terminus of CCDC96 with confidence.

We now change the abstract to be more specific to reflect our finding.

“We also found that the CCDC96/113 complex is in close contact with DRC9/10 in the linker domain.”

The last line of the abstract states that the N-DRC is associated with a network of coiled-coil proteins that may regulate the N-DRC. How does the information in this manuscript support this point? It seems to be a hypothesis that is never tested.

It is true that the hypothesis is not tested in our manuscript. On the other hand, the connection between the N-DRC and the coiled coil networks allows a reasonable explanation of how the N-DRC, as a regulatory hub, can regulate single headed dyneins that might be quite far away from the N-DRC within the 96-nm repeat.

Therefore, we believed that this hypothesis is of high importance and to be tested in the future. While the hypothesis is not directly tested, the loss of some of these coiled coil proteins such as CFAP57, CCDC96/113 and CCDC39/40 negatively affects the regulation of dynein arms (based on previous studies). This inferred the regulatory effects of these coiled coils.

Lines 69-71. The authors state that loss of the DRC proteins rescues paralysis of mutants lacking the radial spokes and central apparatus. Based on the references they cite, these drc mutants suppress mutants that have lost the radial spokes. None of the references report suppression of the central apparatus mutants. Not all drc mutant suppress. Line 69. DRC should be in italics as the authors are discussing a gene and not the protein. This suggests that the authors are not familiar with the literature on this structure in cilia.

The suppressor mutants *sup_pf3 (drc4)* and *sup_pf4 (drc5)* which suppress radial spoke mutations, while *sup_pf5 (drc2)* suppresses central apparatus mutations (Piperno, Mead & Shestak, JCB, 1992). The reference for *sup_pf5* is cited as reference 16 in our manuscript.

We also carefully check all the genes/mutants and italicize them.

In Figure 6, the authors model human variants. Nonsense mutations are usually associated with a reduction in the steady-state level of cytoplasmic mRNA. This mechanism of nonsense mediated decay (NMD) is responsible for the degradation of mRNAs that contain a premature termination codon at a position at least 50 nucleotides upstream of an exon–exon boundary but it is not universal. Since most of the human variants do not have mRNA or protein data, modeling of nonsense variants does not make sense to learn about biological function. It is likely that the premature termination codons result in null alleles and not truncated proteins. Unless there is mRNA or protein data to support the variants in Figure 6A, I would suggest deleting this part of the figure. The missense variants in Figure 6B in DRC2 and DRC4 could be useful. It is possible that the variants destabilize the protein. Is there antibody or proteomics to show that the protein is present in the cilia or in the cell body in these variants? Does the patient phenotypes come from the loss of the protein or from a change in the interaction? Again, it is important to know if the protein is present or just destabilized and lost.

Figure 6A deleted.

Figure 6B is so small that it does not provide any useful information. If Panel A is deleted then, each of the missense variants could be made bigger and the consequence of the missense variant shown by comparing the wild-type and variant proteins using AlphaFold.

We deleted the part related to truncated mutations from the text. Panels B and C are made bigger for better visualization.

In this case, AlphaFold does not predict the disruption of the structures using the missense mutation. However, it is not easy to use AlphaFold to do this task. This is even difficult to verify using in vitro assay because the mutations might be important during the assembly process, not the final structure. What we tried to achieve in this part of the manuscript is to illustrate the usage of a model structure from *Tetrahymena* allowing us to visualize the mutations in human and potentially help interpretation of certain mutations.

Lines 233-280:

The interaction of DRC3 with dynein g was reported in 2015. Interactions with dynein e have previously been reported. Although there is increased resolution, the authors should acknowledge that other methods have already found many of these interactions.

We added the citation for interaction of DRC3 with dynein g (Iwata et al., 2015, PMID: 26063732). For dynein e with N-DRC, we cited Bower et al., 2018, PMID: 29167384.

Discussions of crosstalk and signaling are better for a discussion than the in the results. There are no experimental results about this proposed signalling.

We divided this text into two parts. We put one half in discussion and one part in main text.

Main text:

As suggested by previous studies [25, 36] our result also showed that the N-termini of DRC9 and DRC3 interact with dynein e and g, respectively, suggesting that they are regulatory hubs for such dynein (Fig. 3C). Our subtomogram average of the 96-nm repeat indicates that the CCDC96/113 complex interacts with the base part of RS3 (Fig. 3B), as also suggested by a previous study [12].

Discussion:

The apparent proximity of N-DRC and CCDC96 suggests that CCDC96 has crosstalk with the N-DRC. Therefore, the CCDC96/113 complex may function as a signal transfer bridge for the N-DRC that runs from radial spoke RS3 to DRC10 and vice versa. Regulatory signals can be sent to dynein e through the N-terminus of DRC9. In addition, DRC10 may send signals to dynein g through DRC3. Therefore, the DRC9/10 coiled coil may be the most important regulatory hub in the N-DRC that regulates dyneins g and e by receiving the signal from CCDC96/113.

Figure 3 discusses the protein FAP337. Crosslinking shows interaction with CCDC96. FAP337 is a protein that is reduced in the *Chlamydomonas* mutant *fap57* while CCDC96 is not missing in the *fap57* mutant. Do the authors see evidence of FAP57 or FBB7 (a paralog) in their structures or in their cross-linking experiments?

Yes, we localized FAP57. In the reviewed manuscript, CFAP57 is colored in Figure 3B and stated in the Figure legend. In the text, we also said that “This density extends through the CCDC96/113 complex and interacts directly with the WD40 domain of CFAP337. In *C. reinhardtii*, knockout of CFAP57 leads to loss of CFAP337 [38], suggesting that CFAP57 is a strong candidate for the A3A4 coiled coil.”

Our localization is confirmed by a study using *Chlamydomonas* and human (Walton et al., 2023).

We now make this clear by labelled CFAP57 in the figure 5 and also put the structures of both N-terminal and C-terminal regions of CFAP57 dimer in our model as well.

Dynein d and g are reduced in the fap57 mutant as well. If there is a network, these results suggest the network is not reciprocal.

The structure of CFAP57 (or A3A4 coiled coiled) clearly suggest that it C-terminal domains interact with dynein d and g. So our manuscript actually validate the biochemical results of CFAP57 and dynein d and g interaction.

Minor points

What is a linker part? This term is used repeatedly and I do not know what is means. It is on line 81, 103, 110, 164, 180, 192 and many more. If it is be used, it should be defined.

Linker domain and baseplate domain introduced in the introduction and in figure 1C. We are now make sure to be consistent with our language throughout the paper.

In Figure 1C, the grey is not identified. Please include an explanation. The text states that 85% of the analyzed density is explained; is this the grey regions?. Are these parts that could not be identified with respect to the tomography images of the N-DRC?

The colors used for AlphaFold2_CrossLink and Coot are hard to tell apart. I am not sure if there are two different colors in the cleft or just one.

In Figure 1, the colour of the baseplate domain changed from dark red to yellow for better visualization and better clarification of the method. The gray colour in the linker domain indicates the unidentified densities.

We added the methods based on colour in figure1 legend.

(C) The composite cryo-EM map of N-DRC and DMT. The colors of different regions indicate the modeling methods. Yellow indicates the baseplate part which modeled using Coot. The cyan, red, and green represents the regions modeled using AlphaFold, AlphaFold and Crosslink, AlphaFold and Assemblin, respectively. The remaining gray color in linker domain indicates unidentified regions. Signs (+) and (-) indicate the distal and proximal ends of the DMT.

Line 45: Is there any evidence that motile cilia affect the development and maintenance of the spinal cord in humans? Reiter and Leroux suggest a low frequency of headaches and hydrocephaly.

Motile cilia of ependymal cells are necessary for controlling directional flow of cerebrospinal fluid (CSF) within the ventricular zone (Ref 1-3). This fluid flow plays a crucial role in distributing signaling molecules and nutrients, as well as removing waste products.

Motile cilia-driven fluid flow helps orchestrate the orderly division and migration of neural progenitor cells, ensuring their proper development and organization within the spinal cord (ref 4).

1. Hyland RM, Brody SL. Impact of Motile Ciliopathies on Human Development and Clinical Consequences in the Newborn. *Cells*. 2021 Dec 31;11(1):125.
2. Lehtinen MK et al. 2011 The cerebrospinal fluid provides a proliferative niche for neural progenitor cells. *Neuron* 69, 893-905
3. Kumar V, Umair Z, Kumar S, Goutam RS, Park S, Kim J. The regulatory roles of motile cilia in CSF circulation and hydrocephalus. *Fluids Barriers CNS*. 2021 Jul 7;18(1):31.
4. Cilia in the nervous system: linking cilia function and neurodevelopmental disorders. *Curr Opin Neurol*. 2011 Apr;24(2):98-105.

Line 50: central pair. The field is moving towards calling the two singlet microtubules the central apparatus since it is much more than the pair of microtubules. It is likely to have over 70 proteins and many associated structures. Please consider using central apparatus and not central pair.

We changed all the text to central apparatus.

Line 58: The number of dyneins is organism-specific. There are two outer dyneins in humans but three in *Chlamydomonas* and *Tetrahymena*. There are six inner dynein arms in *Chlamydomonas*, there are reported to be.

We are not clear previously. We revised the text to be clear.

“Each 96-nm repeating unit contains four units of ODAs (two or three-headed depending on the organism), one two-headed IDA (dynein II/f), and six distinct single-headed IDAs (dyneins a-g) with different mechanical properties [3].”

Line 134-136: It is interesting that there are paralogs of DRC4. How different are DRC4A and DRC4B? Are the difference in regions that do not interact in the structure?

Our data (AlphaFold and pulldown) suggested DRC4A and DRC4B form a heterodimer in the cilia (Fig S5A, B).

Interestingly, these two proteins are different in sequence with an identity score of 25.39.

The sequence alignment of two copies of DRC4 is shown below.

aminoacid 185 to 220 of DRC4A interact with DRC3. It seems these residues are rich in E (Glutamic acid).

tr Q23YW7 Q23YW7_TETTS	-----MPP---KKAKGKKKKE---EEPDEYKSMTGADLTQTLEKLERVNE MRT	44
tr I7LT80 I7LT80_TETTS	MSKAAKAKKNVPVVS KLEADARKAAEGVNDNESEEFKKAMRKE----ARALVEQFNEEKK	56
	:* :*:* * : : : : * : : : * : * : *	
tr Q23YW7 Q23YW7_TETTS	NRNYIQMDRDMVENFYHNTLKEISEVKT KISNKETEAEKESKHRIDVKVFLQKVKHLEY	104
tr I7LT80 I7LT80_TETTS	LLAFYQQRQKINYNWIIAKKELEDKSELINKEREIQDLQENHFMTLNQVYKQKIKHLLF	116
	: * : * : : : : : * : : * : : * : * : * : : * : : * : * : * : * : *	
tr Q23YW7 Q23YW7_TETTS	EQEKSNLNIEDDGKKAKEKEDAYFEDI--TKNMKQLKTQLKSEYLEKEKANIQ---QVQEE	160
tr I7LT80 I7LT80_TETTS	QNQDQSEELKKDVEVTLKQ---LEDQHRIKSRLEKTDVRS LKVTKKEQEISQQDYLFAL	172
	: : : : : : * : : : : * : : : * : : * : : * : * : * : * : * : * : *	
tr Q23YW7 Q23YW7_TETTS	KKDHQSLLKIQQKFPDELINNLIIKYEERLAKLKEDELEKLVKVEIHELEERKNLHINELM	220
tr I7LT80 I7LT80_TETTS	KSEHDKQMTLMRQDYERQVNDIKRKYDLKMQLNTRMEEARAAMIKQLEDNKNQKIAEII	232
	*. : * : : : : : : * : : * : : : * : : * : * : * : * : * : * : * : *	
tr Q23YW7 Q23YW7_TETTS	NNHEKAFAPLKKYYNDITAENLNLIKAKHKEKIAQIYANIQLNNTKNVADNQAKNEQLKEPL	280
tr I7LT80 I7LT80_TETTS	KEHTQKYNDIKNYSEITATNLDYRKT LKNEIKELQTKDEEYKKTQQTEKGYKELNEPL	292
	: * : : : * : * : * : * : * : * : * : * : * : * : * : * : * : * : * : *	
tr Q23YW7 Q23YW7_TETTS	AKHREIRNKLKEDLKQFAKHKMSLQNLKSKAITLKDKITKLERDGGKDLDEKYE-----	333
tr I7LT80 I7LT80_TETTS	QALGI-----EIIQLKKQDEEQEAIKEKEELKQKIDNQERLFRKLEYEYEVKLQQFQ	345
	: : * : * : . : : : : : * : * : * : * : * : * : * : * : * : * : * : *	
tr Q23YW7 Q23YW7_TETTS	KVVREKQELEKKFEDITQEVKKNADLNNNVLSNR LQILLKEYNNKEEELRTIIDNAGLDH	393
tr I7LT80 I7LT80_TETTS	YLERERNALYAKFNQTVFEIHQKSGLENLILEKKVTNLRDELEIKDLQIHQVLTAAIDP	405
	: * : : * * : : . * : : : : * : * : * : * : * : * : * : * : * : * : * : *	
tr Q23YW7 Q23YW7_TETTS	NLHEQLQRVQQSIEAKNTLIKNLKYSIHHATKAYNDAIRVYEAQLVDFGPIEELGFQP	453
tr I7LT80 I7LT80_TETTS	NSVGSINKSLEEVESLKNELISELQAQLKIRKAHSHMVKAYEGKLSFVIVPEELGFDP	465
	* . : : : : : . * * * : * : : : * : : . : : * : * : * * * : * : * : * : *	
tr Q23YW7 Q23YW7_TETTS	LETITSSMPAGLVSS	468
tr I7LT80 I7LT80_TETTS	LVPTNTD-----	472
	* . : .	

Line 176: The original study of FAP91 was by Dymek, Heuser, Nicastro and Smith (2011) and this reference should be cited.

Reference added.

Line 268: Loss of proteins in a mutant suggests that the assembly requires the mutant protein but does not show interaction.

We revised our text to the following to be clear.

“Mass spectrometry analysis confirmed that both CFAP337A and CFAP337B are lost in CCDC96 knockout cells [12], suggesting that CFAP337A/B interacts with CCDC96.”

DRC12 is reported to be an N-DRC protein. Chlamydomonas has no ortholog and proteomics of isolated airway cilia do not show its presence. Is there any data to support this protein in cilia or flagella?

Chlamydomonas’s ortholog of DRC12 is FAP405 (Chlamyfp.org). FAP405 only exists since in v5 of Phytozome database. It explains that no MS study before 2018 detected this protein. According to Chlamyfp.org’s human cilia proteins, CCDC153 is listed as a ciliary protein

(http://chlamyfp.org/ChlamyFPv2/hs_read_sql.php). Ciliacarta also suggests CCDC153 is a ciliary protein (<https://tbb.bio.uu.nl/john/syscilia/ciliacarta/>).

For the isolated airway cilia proteomics, we assumed the publication mentioned is Blackburn et al. 2017 (<https://pubmed.ncbi.nlm.nih.gov/28282151/>).

Since the paper was published a while ago, the coverage of the proteins in the paper is certainly not as good (> 400 proteins). For example, among all the DRC proteins (Supplementary Table 1), the study detected DRC1, DRC2, DRC3, DRC4, DRC5, DRC7, DRC10 but not DRC6, DRC8 and DRC9. DRC12 is a small protein, of similar size to DRC8. Therefore, it is not unexpected to not detect DRC12 in the isolated airway cilia proteome.

Reviewers' Comments:

Reviewer #1:

Remarks to the Author:

The new manuscript was significantly improved according to the reviewers' comments. Especially the new "Table S1 helps readers to understand the reliability of each protein model. Because the resolutions of cryo-EM maps significantly differ between the base plate region and the linker region due to the flexibility of the N-DRC structure, the authors combine structural and biochemical approaches to assign the protein models. The table comprehensively summarizes the rationale for each model's assignment in this study.

Moreover, according to the reviewer's comment, the authors added Figures S7C and 5E, which show the comparisons between the cryo-EM model (isolated DMT) and the cryo-ET structure (intact cilium). The interface structure between the N-DRC and the adjacent DMT remains elusive. These figures suggest the molecular basis of the N-DRC-PF B8 connection and the stressed conformation of the N-DRC linker in the intact cilium, which might be critical for constructing the whole axoneme.

However, this manuscript contains several misprints related to the revisions for the DRC6C and Calmodulin identification. The authors themselves should have revised these misprints before resubmission of this manuscript. They diminish the reliability of the whole data and statement in the manuscript.

Specific comments are as follows:

(1)

Figure 3 should be replaced with the correct one. Figure 3 in this manuscript is the same as the previous version of Figure 2.

(2)

In Figure 2B, a model colored gray should be annotated like other models. This gray model seems to show the DRC6A instead of DRC6C. On the other hand, Figure 1D uses the DRC6C model as the DRC6 structure. Why do the authors use different models between Figures 1D and 2B? If this differs from the misprint, please clarify the reason.

(3)

For the validation of the assignments of DRC6C and CaM models, the authors should generate a figure which shows the cross-linking sites of DRC12-DRC6C and DRC6C-CaM. Figure S2B summarizes the inter and intra-molecular cross-linking sites. However, this figure lacks CaM and uses DRC6A instead of DRC6C. Why do the authors avoid applying the inter-molecular cross-linking information of DRC6C and CaM to Figure S2B? If this differs from the misprint, please clarify the reason.

(4)

The reviewer speculates that Figure 2D was generated based on the model in Figure 1D. The Figure 1D model has DRC6C and CaM at the top of the N-DRC linker. On the other hand, Figure 2D annotates DRC6A in the legend (line 883). Why do the authors refer to different molecules (DRC6A and DRC6C) between Figures 1D and 2D? Again, if this differs from the misprint, please clarify the reason.

(5)

The red lines in Figures 4B and 4D are so large that they mask the interaction sites between N-DRC and IDA. Please remove or modify them.

(6)

For the comparison of DRC6 structures, the authors should add DRC6B and DRC6C models in Figure S3B.

Reviewer #2:

Remarks to the Author:

The authors have made very useful corrections and comments to the reviews. I am satisfied. I still am wary about DRC12 as it quite divergent. When CCDC153 is compared to the Chlamydomonas genome, this protein has an E value of $4.28e-3$. Their other data suggest it is a part of the N-DRC. This is very nice and significant work.

First, I would like to thank the reviewers for helpful suggestions and comments. We apologise for some of the misprints due to a rush from overlapping vacations and short revision time. On the other hand, we are confident with everything we included in the last revision. This time, we carefully looked at everything and addressed the following issues:

- Corrected all the minor mistakes mentioned in the review and more in text, figure captions.
- Thoroughly looked at every figure, supplementary figures and tables to update with our newest model.
- Updated the tomography workflow and figure (Supplementary Figure 6).
- Included the source data according to the guideline.

REVIEWERS' COMMENTS

Reviewer #1 (Remarks to the Author):

The new manuscript was significantly improved according to the reviewers' comments. Especially the new "Table S1 helps readers to understand the reliability of each protein model. Because the resolutions of cryo-EM maps significantly differ between the base plate region and the linker region due to the flexibility of the N-DRC structure, the authors combine structural and biochemical approaches to assign the protein models. The table comprehensively summarizes the rationale for each model's assignment in this study.

Moreover, according to the reviewer's comment, the authors added Figures S7C and 5E, which show the comparisons between the cryo-EM model (isolated DMT) and the cryo-ET structure (intact cilium). The interface structure between the N-DRC and the adjacent DMT remains elusive. These figures suggest the molecular basis of the N-DRC-PF B8 connection and the stressed conformation of the N-DRC linker in the intact cilium, which might be critical for constructing the whole axoneme.

However, this manuscript contains several misprints related to the revisions for the DRC6C and Calmodulin identification. The authors themselves should have revised these misprints before resubmission of this manuscript. They diminish the reliability of the whole data and statement in the manuscript.

Specific comments are as follows:

(1)

Figure 3 should be replaced with the correct one. Figure 3 in this manuscript is the same as the previous version of Figure 2.

We uploaded the wrong figure last time. Corrected.

(2)

In Figure 2B, a model colored gray should be annotated like other models. This gray model seems to show the DRC6A instead of DRC6C. On the other hand, Figure 1D uses the DRC6C model as the DRC6 structure. Why do the authors use different models between Figures 1D and 2B? If this differs from the misprint, please clarify the reason.

Figure 1D and 2B are corrected.

(3)

For the validation of the assignments of DRC6C and CaM models, the authors should generate a figure which shows the cross-linking sites of DRC12-DRC6C and DRC6C-CaM. Figure S2B summarizes the inter and intra-molecular cross-linking sites. However, this figure lacks CaM and uses DRC6A instead of DRC6C. Why do the authors avoid applying the inter-molecular cross-linking information of DRC6C and CaM to Figure S2B? If this differs from the misprint, please clarify the reason.

DRC6C and CaM and crosslinks are added to Figure S2B.

(4)

The reviewer speculates that Figure 2D was generated based on the model in Figure 1D. The Figure 1D model has DRC6C and CaM at the top of the N-DRC linker. On the other hand, Figure 2D annotates DRC6A in the legend (line 883). Why do the authors refer to different molecules (DRC6A and DRC6C) between Figures 1D and 2D? Again, if this differs from the misprint, please clarify the reason.

The legends are modified and DRC6A is changed to DRC6C.

(5)

The red lines in Figures 4B and 4D are so large that they mask the interaction sites between N-DRC and IDA. Please remove or modify them.

The red lines are fixed in Figure 4D and 4B.

(6)

For the comparison of DRC6 structures, the authors should add DRC6B and DRC6C models in Figure S3B.

DRC6B and DRC6C are added to Figure S3B.

Reviewer #2 (Remarks to the Author):

The authors have made very useful corrections and comments to the reviews. I am satisfied. I still am wary about DRC12 as it quite divergent. When CCDC153 is compared to the Chlamydomonas genome, this protein has an E value of $4.28e-3$. Their other data suggest it is a part of the N-DRC. This is very nice and significant work.

Thank you.